# Plants as a Source of Anticancer Agents: From Bench to Bedside

**DOI:** 10.3390/molecules27154818

**Published:** 2022-07-27

**Authors:** Wamidh H. Talib, Safa Daoud, Asma Ismail Mahmod, Reem Ali Hamed, Dima Awajan, Sara Feras Abuarab, Lena Hisham Odeh, Samar Khater, Lina T. Al Kury

**Affiliations:** 1Department of Clinical Pharmacy and Therapeutic, Applied Science Private University, Amman 11931, Jordan; asmamahmod1212@gmail.com (A.I.M.); r_hamed@asu.edu.jo (R.A.H.); d_awajan@asu.edu.jo (D.A.); saraarab97@yahoo.com (S.F.A.); lena_odeh@hotmail.com (L.H.O.); samar_ktr@asu.edu.jo (S.K.); 2Department Pharmaceutical Chemistry and Pharmacognosy, Faculty of Pharmacy, Applied Science Private University, Amman 11931, Jordan; s_daoud@asu.edu.jo; 3Department of Health Sciences, College of Natural and Health Sciences, Zayed University, Abu Dhabi 144534, United Arab Emirates; lina.alkury@zu.ac.ae

**Keywords:** plant-derived natural products, alternative cancer therapies, anticancer, apoptosis induction, plants

## Abstract

Cancer is the second leading cause of death after cardiovascular diseases. Conventional anticancer therapies are associated with lack of selectivity and serious side effects. Cancer hallmarks are biological capabilities acquired by cancer cells during neoplastic transformation. Targeting multiple cancer hallmarks is a promising strategy to treat cancer. The diversity in chemical structure and the relatively low toxicity make plant-derived natural products a promising source for the development of new and more effective anticancer therapies that have the capacity to target multiple hallmarks in cancer. In this review, we discussed the anticancer activities of ten natural products extracted from plants. The majority of these products inhibit cancer by targeting multiple cancer hallmarks, and many of these chemicals have reached clinical applications. Studies discussed in this review provide a solid ground for researchers and physicians to design more effective combination anticancer therapies using plant-derived natural products.

## 1. Introduction

Cancer is a chronic disease and one of the main causes of death around the world. The global cancer burden is extensively rising, and it is considered as the second cause of death after cardiovascular disease. In 2020, 19.3 million new cancer cases were estimated, with 10.0 million cancer deaths worldwide and 250 million disability-adjusted life years because of cancer [1,2]. Regardless of the new developments in cancer therapy and revolutionary advances in genomics and molecular biology, multidrug resistance and drug side effects are still the vital cause of cancer treatment failure [3]. Since plants are a rich source of natural compounds that are characterized by their therapeutic effects, studying these compounds is thought to be a promising line for research on cancer [4]. In this context, phytochemicals, secondary metabolites extracted from plants, have diverse applications, including antidiabetic, anti-inflammatory, cardiovascular protective, antioxidant, and anticancer effects [5]. In particular, these phytochemicals can be classified into different groups such as flavonoids, alkaloids, phytosterols, terpenoids, sulfides, polyphenols, and others, which have been considered an important reservoir for novel anticancer agents [6,7,8,9]. Hence, plant secondary metabolites are recognized with many properties such as tumor growth inhibition, apoptosis induction, immune modulation, and angiogenesis suppression [4,10]. As well, several epidemiological studies have reported the role of phytochemicals and their derived analogues in modulating tumor cell-activating proteins, enzymes, and signaling pathways, stimulating DNA repair mechanisms, and conquering free radicals production [4,11,12]. They also interact with many intracellular pathways that regulate cell growth, such as the STAT3, PI3K/Akt/NF-κB signaling pathway, mTOR, and the Bcl-2/Bax mitochondrial pathway [9,13].

In this review, we have tried to choose the most effective and well-known phytochemicals that display a distinctive anticancer activity. As well, we described these phytochemicals thoroughly, starting with their chemical structure and ending with their antitumor activity.

## 2. Cancer Hallmarks as Targets for Natural Products

Cancer hallmarks are biological capabilities that are acquired during neoplastic transformation and help in organizing the complexity of cancer development. Many natural products exert their anticancer effect through targeting one or multiple cancer hallmarks, which are briefly discussed below.

### 2.1. Genomic Instability

Genomic instability is a property of many tumor cells that can be triggered by different mechanisms such as telomerase destruction, centrosome amplification, epigenetic modifications, and DNA impairment [14]. In a normal cell cycle, the integrity of the genome is controlled by specific checkpoints and any abnormality in function of these checkpoints can lead to development of tumorigenesis [15,16]. These checkpoints are regulated by different oncogenes and tumor suppressor genes; nevertheless, cancer cells can modify the functions of these genes resulting in stimulation of uncontrolled cell growth [16].

### 2.2. Sustained Proliferative Signaling

Modulating the expression of growth-promoting signals is a prominent feature of cancer cells in order to maintain their uncontrolled cell division and proliferation [17]. The essential targets to inhibit sustained proliferation in cancer include the following signaling pathways: hypoxia-inducible factor-1 (HIF-1), NF-κBs, PI3K/AKT, insulin-like growth factor receptor (IGF-1R), cyclin-dependent kinase (CDKs), and estrogen receptor signaling [18].

### 2.3. Evasion of Anti-Growth Signaling

The evasion of antigrowth signals is another strategy implemented by cancer cells to preserve proliferation. Blocking tumor suppressor genes that regulat the antigrowth signals, as well as mutations in these genes, have been detected in cancer cells [16]. In addition, the most identified mutated tumor suppressor genes is p53 followed by ataxia-telangiectasia mutated (ATM), cyclin-dependent kinase inhibitor 2A (CDKN2A), phosphatase and tensin homolog (PTEN), adenomatous polyposis coli (APC), breast cancer gene 1 and 2 (BRCA1 and BRCA2), retinoblastoma (RB), and Wilms tumor (WT1) [18].

### 2.4. Resistance to Apoptosis

Cancer cells can promote the overexpression of anti-apoptotic proteins eventually suppressing the normal programmed cell death [17]. Moreover, cancer cells can limit or bypass apoptosis via many pathways such as altering the function of p53 tumor suppressor gene, increasing the expression of antiapoptotic regulators (Bcl-2 and Bcl-xL), promoting survival signals (Igf1/2), reducing the levels of proapoptotic factors (Bax, Bim, Puma), and suppressing the signals of the extrinsic ligand-induced death pathway [17].

### 2.5. Replicative Immortality

Cancer cells are recognized for their limitless replicative potential, which mediates tumor progression and invasion. Telomerase is a specialized reverse transcriptase that extends the ends of shortening chromosomes in dividing cells [16,19]; hence, activation of this enzyme is the key to maintain continuous cell division in many types of cancer [17]. In addition, replicative immortality can be modulated by suppressing many targets including telomerase, mTOR, CDK4/6, CDK 1,2,5,9, Akt, and PI3K [18].

### 2.6. Dysregulated Metabolism

Altering energy metabolism has been confirmed to be a cancer-associated trait, which involves the stimulation of many oncogenes and mutated suppressor genes [17]. In order to increase glucose uptake and lactate production, several glycolytic enzymes are activated, including hexokinase 2 (HK2), 6-phosphofructo-2-kinase/fructose-2,6-biphosphatase 3 (PFKFB3), and pyruvate kinase isoform M2 (PKM2) [16]. In addition, overexpression of other metabolic regulators such as hypoxia-inducible factor 1 (HIF-1) and Myc oncogene was observed in cancer cells [20,21].

### 2.7. Tumor-Promoting Inflammation

A specific association between chronic inflammation and cancer development has been observed [22,23]. Furthermore, several factors have been identified for their crucial role in stimulating cancer-related inflammation, including cytokines (interleukins, TNF-α, TGF-β, and granulocyte macrophage colony-stimulating factor), chemokines, and transcription factors (NF-kB, STAT3, HIF-1-α) [22].

### 2.8. Angiogenesis

During the malignancy stage, an “angiogenic switch” is triggered in tumor cells, which involves stimulation of angiogenic factors that mediate vascularization (blood vessels formation) [24]. These new growing blood vessels would supply the dividing cancer cells with oxygen and nutrients, which are essential to sustain cell proliferation [24]. This angiogenesis process is regulated by many transmembrane proteins and pathways, including vascular endothelial growth factor (VEGF), VEGF receptor 2 (VEGFR2), Tie-angiopoietin pathways, platelet-derived growth factor (PDGF), epidermal growth factor (EGF), and hepatocyte growth factor (HGF) [24,25].

### 2.9. Tissue Invasion and Metastasis

Cancer metastasis is a multi-step process that starts with local invasion of cancer cells into the surrounding tissues. The next step is intravasation into the nearby vessels, extravasation to distant tissues and organs, and finally adaptation to a new microenvironment during which micro-metastases will progress into a secondary tumor [17]. Tumor cell metastasis is known to be initiated by the disruption of cell–cell adhesion, which is comprised of tight junctions, adherens junctions, gap junctions, desmosomes, and hemidesmosomes [26].

### 2.10. Immune Evasion

Cancer cells apply different strategies to evade immune surveillance. These include modulating immune checkpoint pathways and recruiting immunosuppressive cells (e.g. regulatory T cells and myeloid-derived suppressor cells). Additional mechanisms involve impairing some elements of the immune system (e.g. suppressing infiltrating CTLs and NK cells by overexpression of TGF-β or other immunosuppressive factors) [17]. Figure 1 summarizes the cancer hallmarks with the main regulating markers.

## 3. Anticancer Drug Discovery

For decades, plants have been well known for their clinical usefulness as anticancer agents using diverse mechanisms of action [27,28]. Mostly, this is due to their abundant quantity of secondary metabolites, which possess significant pharmacological and biological activities including antitumor activity [29]. About 50% of antitumor drugs originate from natural products. These drugs could be isolated from plants or semi-synthetic compounds [30]. Such examples of antitumor drugs of clinical importance nowadays are taxanes (e.g. Taxol), vinblastine, vincristine, and the podophyllotoxin analogs that all originate from higher plants [31]. Firstly, in the 1950s, Robert Noble and Charles Beer of Canada discovered vincristine and vinblastine vinca alkaloids, which were isolated from the leaves of *Catharanthus roseus* (Madagascar periwinkle) [32]. In addition, taxane-derived drugs are isolated from the *Taxus* genus. Paclitaxel, which is commercially identified as Taxol^®^, was firstly obtained in 1971 from *Taxus brevifolia* (Pacific yew). In the 1980s, another taxane semi-synthetic drug was isolated from *Taxus baccata* (European yew) known as docetaxel (Taxotere^®^) [33]. Recently, podophyllotoxin analogs have been of great interest in chemical modification, especially the semi-synthetic antitumor drugs teniposide and etoposide [34]. Podophyllotoxin was first obtained by Podwyssotzki in 1880 from the North American mayapple *Podophyllum peltatum* L. Moreover, this phytochemical has been collected from *Podophyllum emodi* (*Indian podophyllum*) [35]. Etoposide was synthesized for the first time in 1966 and was approved by the U.S. Food and Drug Administration (FDA) for cancer therapy in 1983 [36].

## 4. Chemoprevention Mechanisms of Plant-Derived Products

Cancer chemoprevention is a comprehensive term that defines the contribution of exterior agents to suppress cancer development. Many plants play a vital role in the chemoprevention process depending on various mechanisms [37]. Recently, researchers have paid attention to natural products as chemopreventive agents due to their low toxicity, availability, and affordable production cost [38]. In this review, we will cover the mechanisms of chemoprevention of the already-included natural products. Starting with curcumin, it has been found that it exerted a chemoprevention activity on ovarian cancer. This activity is carried out by inhibiting the NF-κB signaling pathway [39]. Needless to say, the inhibition of NF-κB is crucial in cancer treatment and prevention [40]. In addition, curcumin and its analog and metabolite have an antimetastatic activity by decreasing Hsp70 and toll-like receptor (TLR4). Moreover, it induces apoptosis via increasing caspase-9 and caspase-3 [41]. Resveratrol stimulates mitogen-activated protein (MAP) kinase phosphatase-1 (MKP-1) [42] and nuclear factor erythroid 2-related factor 2(Nrf2) [43]. Moreover, resveratrol suppresses NLRP3 inflammasome and cyclooxygenase (COX) [44,45,46]. Moreover, quercetin is widely known as an antioxidant and cell protective agent. It could scavenge reactive oxygen species (ROS) and decrease the risk of cancer development and DNA damage as well [47]. Quercetin activates apoptosis through augmented translocation of Bax to the mitochondrial membrane, rearrangement of Bcl-2 proteins, stimulation of caspases, and blockage of the ERK and PI3K/Akt signals [47,48,49,50]. Furthermore, EGCG has been confirmed to possess a chemopreventive activity via the suppression of carcinogenesis processes such as initiation, progression, and promotion [51]. EGCG showed remarkable regulatory effects on several signaling pathways such as NF-κB, JAK/STAT, AP-1, MAPK, and PI3K/AK [52,53,54,55,56,57,58,59]. Interestingly, EGCG has the ability to control the activity of androgen receptors in prostate cancer as well as the estrogen receptors in mammary cancers [60,61]. In A549 and H1299 lung cancer cell lines, allicin has proven its antineoplastic activity by modification of PI3K/AKT signaling. This modulation leads to the suppression of cellular invasion, proliferation, and metastasis [62,63]. Bat-Chen et al. demonstrated that allicin mediates apoptosis via induction of Nrf2, increased levels of hypodiploid DNA content, enhanced Bax levels, and decreased B-cell non-Hodgkin lymphoma-2 levels (Bcl-2) in human colon cancer HCT-116 cells [64]. Thymoquinone inhibits cell proliferation via taking control over main cell cycle checkpoints and encouraging cell cycle arrest at G1/G2, G0/G1, or G2/M phases [65]. TQ therapy remarkably reduces angiogenesis. Thus, through the extracellular receptor kinases pathway and Akt, TQ modulates the signal of vascular endothelial growth factor (VEGF) [66]. It acts as an antioxidant as well and augments the levels of the antioxidant HO-1, Nrf2, and SOD proteins [67]. Emodin leads to apoptosis induction and epithelial to mesenchymal transition (EMT) inhibition in colon cancer [68,69,70]. This phytochemical inhibits endothelial cell proliferation via cell cycle arrest in the G2/M phase. Similarly, it inhibits VEGFR2 and MMPs signaling pathways. Moreover, emodin has been defined as an inhibitor of tyrosine kinase and phosphorylation of ERK 1/2 downregulator, contributing to its antiangiogenic effect [71,72,73]. Genistein inhibits hepatocellular carcinoma in rats by suppression of the PDGF/versican bidirectional axis, inhibiting both PKC and ERK1 as downstream controllers [74]. In ovarian cancer, it leads to serum malondialdehyde downregulation, which is a marker for the expression of Bcl-2 and NFκB and oxidative stress as well. On the other hand, it upregulates HO-1, Nrf2, and Bax expression in ovarian tissues [75]. Genistein causes PTK signaling blocking by inhibition of protein-tyrosine kinase (PTK) and thus suppresses the proliferation of tumor cells indirectly [76]. In esophageal carcinoma, parthenolide (PTL) has been found to suppress the NF-кB/AP-1/VEGF signaling pathway [77]. Moreover, PTL has been revealed to induce classic apoptosis, as well as alterations in the Bcl-2 family and rising reactive oxygen species (ROS) production intracellularly [78]. PTL exerts proapoptotic stimulation of p53, together with reduced glutathione reduction [79]. Luteolin led to pancreatic cancer inhibition in a BOP-induced hamster model. This was achieved by downregulation of the STAT3 dihydropyrimidine dehydrogenase (DPYD) pathway, which has a central importance in the development of pancreatic cancer [80]. Additionally, luteolin activates apoptosis, suppresses proliferation, and modifies MAP kinase and Akt/mTOR pathways in HeLa human cervical cancer cells [81].

## 5. Plant-Derived Natural Products with Potential Anticancer Effects

### 5.1. Curcumin

Curcumin, also called diferuloylmethane, is the main natural polyphenol found in the rhizome of *Curcuma longa* (Family: Zingiberaceae) and in multiple other *Curcuma* species [39]. It is a yellow natural product, commonly known as turmeric, commonly used as a food spice and colorant [82]. The chemical structure of curcumin consists of two aromatic O-methoxy phenolic groups, β-dicarbonyl moiety and a seven-carbon linker containing two enone moieties. The IUPAC name for curcumin is (1E,6E)-1,7-bis(4-hydroxy-3-methoxyphenyl)-1,6-heptadiene-3,5-dione) [83,84].

Curcumin has been traditionally used in Asian countries as a medical herb due to its multiple health benefits [85]. These include anti-inflammatory, antioxidant, antibacterial, antiviral, anti-diabetic, wound-healing ability, and potential anticancer and chemopreventive activity [85,86].

Curcumin has been shown to inhibit carcinogenesis in preclinical studies performed on various cell lines, including prostate, pancreatic, ovarian, oral epithelial leukemia, hepatic, breast, cervical, gastric, and colon cancer [87]. Several mechanisms have been suggested to explain the promising results of curcumin in cancer treatment. One of the major mechanisms of curcumin anticancer effects is its antioxidant properties, since it can increase serum concentration of antioxidants such as superoxide dismutase (SOD), glutathione peroxidase (GPx), and lipid peroxides. Moreover, curcumin also acts as a good scavenger for different forms of free radicals, such as reactive oxygen (ROS) and reactive nitrogen species (RNS), and inhibits ROS-generating enzymes such as lipoxygenase/cyclooxygenase and xanthine hydrogenase/oxidase [88,89,90]. In addition, curcumin can modulate several signaling pathways associated with cancer growth, such as suppression of angiogenesis and induction of apoptosis [91]. Moreover, Yang et al. reported that curcumin also induces apoptosis through the activation of the JNK/ERK/AP1 pathways in human acute monocytic leukemia THP-1 cells [92]. The anticancer effect of curcumin can also be exerted via downregulation of pyruvate kinase M2 (PKM2), hence decreasing lactate production and uptake of glucose in cancer cells [93].

Curcumin can also induce cell cycle arrest at the G2/M phase, probably by decreasing the expression of CDC2 and CDC25, and increasing P21 expression [94]. This result was observed in hematological cancers [95]. In addition, curcumin has affected the cellular development in non-small cell lung cancer and stimulated G0/G1 phase arrest via MTA1 (metastasis-associated protein 1)-mediated deactivation of the Wnt/β-catenin pathway [96]. Additionally, several studies suggest that curcumin exerts its anticancer properties by targeting different microRNA (miRNA) expressions, such as miR-181b, miR-203, miR-9, miR-19, miR-21, miR203, miR-9, and miR-208 [97]. Needless to say, miRNA plays a crucial role in various physiological conditions, including differentiation, growth, angiogenesis, and apoptosis. Hence, dysregulation of these molecules can upregulate and downregulate several cellular and molecular targets leading to the progression of cancer [98]. On the other hand, the synergistic role of curcumin in combination with other chemotherapeutic agents has been reported, an effect that leads to the enhanced effectiveness and decreased toxic effects of these drugs [99]. In another study, Zhang et al. reported that pre-treatment with curcumin followed by 5-fluorouracil increased the susceptibility of the colon cancer cells [100]. Likewise, Guorgui et al. have shown that a combination of curcumin and doxorubicin demonstrates a stronger additive effect by causing a 79% reduction in the proliferation of Hodgkin lymphoma (L-540) cells [101]. Similar synergistic activity of curcumin with docetaxel [102], cisplatin [103,104], and doxorubicin [105] has been also reported.

Interestingly, curcumin has been shown to reverse the multidrug resistance of cancerous cells [106]. Xiao-qing et al. have shown that curcumin can reverse the multidrug resistance of the human gastric carcinoma SGC7901/VCR cell line, thus increasing the sensitivity to vincristine by decreasing both the function and expression of P-gp, resulting in high intracellular drug concentration [107]. Recently, similar findings regarding the potential of curcumin to reverse the multidrug resistance towards doxorubicin in colon cancer treatment have been also reported [108].

On the other hand, curcumin displays no toxicity effect on normal cells [109]. A phase I human trial with 25 patients taking up to 8000 mg of curcumin per day for three months discovered no harm from curcumin [109]. Five further human trials utilizing 1125–2500 mg of curcumin per day revealed it to be safe [109].

### 5.2. Resveratrol

Resveratrol is a polyphenolic compound produced by plants in response to environmental stress and can be found in at least 72 plant species, such as mulberries, peanuts, cranberries, blueberries, and grapes [110]. Resveratrol is a natural stilbene containing two aromatic rings linked together by a methylene double bond to form 3,4′,5-trihydroxystilbene. It exists in both *cis-* and *trans-* isoforms. The *trans* form is more stable and potent than the *cis* form [111]. Studies have reported that resveratrol exhibits many wide ranges of activities, including antioxidant [112], anti-inflammatory [113], cardiovascular protective [114], anti-aging [115], and anticancer properties [116].

Recently, several studies have focused on the anticancer properties of resveratrol, which revealed a high ability to target multiple pathways involved in cancer initiation, promotion, and progression [46,117]. First of all, the antioxidant effects of resveratrol contribute significantly to the health benefits of this phytochemical, since it can act as a scavenger of a number of free radicals. In addition, resveratrol increases the expression of various antioxidant enzymes, such as SOD, catalase (CAT), glutathione reductase (GS-R), (GPx), and glutathione S-transferase (GST). Thus, resveratrol, through these two antioxidant effects, protects cells from oxidative damage [118]. Conversely, as chemotherapeutics, resveratrol in combination with As2O3 can enhance apoptosis-inducing oxidative stress via induction of ROS [119].

A second possible anticancer mechanism of resveratrol is related to kinases, which play a critical role in cell growth and proliferation and are typically over-expressed in many tumors. Resveratrol specifically targets many kinases, including EGF, extracellular-signal-regulated kinases (ERK), and VEGF, thus decreasing their expression and resulting in antigrowth signaling activity [120].

A third suggested mechanism for the anticancer activity of resveratrol is related to its anti-inflammatory activity, since several types of cancer are, to some extent, promoted by a certain degree of systemic, low-grade chronic inflammation [121]. Ren et al. have reported that resveratrol suppresses the inflammatory biomarker tumor necrosis factor α (TNF-α-)-induced signaling in a dose-dependent manner, both via nuclear factor kappa-B (NFκB) activation and transcriptional activity of p65 [122]. Furthermore, in colon cancer, resveratrol actually encouraged cell cycle arrest at the G1 phase. That was achieved by lowering cyclin E1 and cyclin D1 and increasing p53 in a dose-dependent manner [123]. In addition, resveratrol treatment resulted in increasing p27 and p21 gene expression levels, as well as lowering cyclin B gene expression [124]. Moreover, substantial evidence has demonstrated that resveratrol can induce apoptosis in a wide variety of cancer cells, although the underlying mechanism differs greatly among different cancer cell types [46]. Li et al. suggested that resveratrol induces apoptosis through activating caspase-3 and caspase-9, upregulating Bcl-2-associated X protein, and inducing expression of p53 [125]. On the other hand, in ovarian cancer cells, resveratrol has been shown to inhibit proliferation and induce apoptosis via inhibiting glycolysis and targeting the AMPK/mTOR signaling pathway [126]. Moreover, resveratrol can inhibit metastasis by targeting different pathways. In pancreatic cancer cells, resveratrol suppresses the metastatic potential in vitro by modulating EMT-related factors via the PI3K/Akt/NF-κB signaling pathway [127]. Likewise, resveratrol has been shown to inhibit the invasiveness and metastasis of prostate cancer cells by downregulating glioma-associated oncogene homolog 1, which is a transcription factor in the Hedgehog signaling pathway [128]. Recent studies suggested that the antitumor effects of resveratrol can also be mediated through enhancing antitumor immunity and reversing the immunosuppressive tumor microenvironment, which is established via stimulating cytokines/chemokines secretion and expression of several other immune-related genes [129]. These mechanisms were supported by the findings of Lee et al. which demonstrated that resveratrol increases the expression of activating receptors on natural killer (NK) cells, an effect which could facilitate NK-cell-mediated killing of cancer cells [130]. In closing, it remains to be said that, like many other phytochemicals, resveratrol shows a synergistic inhibitory effect on the proliferation of various cancer cells and increases their chemosensitivity to many chemotherapeutic agents, such as temozolomide, cisplatin, doxorubicin, 5-fluorouracil, gemcitabine, docetaxel, and paclitaxel [131].

Lastly, despite the high cytotoxicity of resveratrol toward tumor cells, it appears to be well tolerated with no significant harm recorded against normal cells [132]. These findings are significant in the context of human efficacy investigations, and they lend support to the use of resveratrol as a pharmacological agent in human medicine [132].

### 5.3. Quercetin

Quercetin is a natural lipophilic product and one of the most plentiful flavonoids. Flavonoids are known for their low molecular weight and phenolic structure that mainly exists in the seeds, bark, leaves, and flowers of plants [133]. Flavonoids are categorized into six classes and quercetin falls under the subclass of flavonols.

Quercetin’s name comes from the Latin *quercetum*, meaning oak forest, after the oak genus *Quercus*, and its IUPAC name is 3,3′,4′,5,7-pentahydroxyflavanone (or) 3,3′,4′,5,7-pentahydroxy-2-phenylchromen-4-one, with the molecular formula C15H10O7. Quercetin has three rings with five hydroxyl groups attached at positions 3, 5, 7, 3′, and 4′ that play a major role in its biological activity and formation of derivatives. The main groups of derivatives are glycosides, ethers, and less frequently sulfate and prenyl substituents [134]. Quercetin naturally is an aglycone, missing a glycosyl group, so glycosidic derivatives are achieved by replacing the hydroxyl group (mainly at position 3 and less frequently at position 7) with a sugar-like glucose, rhamnose, or rutinose, resulting in increased water solubility, absorption, and in vivo effects compared to quercetin aglycone [135,136,137]. On the other hand, ether derivatives are achieved by forming an ether bond between any hydroxyl group and an alcohol, mainly methanol [138]. Quercetin is found in many foods such as capers, apples, berries, brassica vegetables, grapes, pepper, asparagus, onions, broccoli, shallots, cherries, tea, and tomatoes. Quercetin is also well observed in medicinal plants, such as *Ginkgo biloba*, *Hypericum perforatum*, and *Sambucus Canadensis* [139,140,141].

Regarding its pharmacological effects, quercetin has been shown to exhibit antiviral, antibacterial, anti-inflammatory as well as anticarcinogenic effects [142,143,144]. The anticarcinogenic effects are mediated through several mechanisms, such as targeting free radicals, inducing apoptosis, regulating cell cycle, and targeting important key molecules in cancer development, in addition to a unique impact on certain types of cancer.

Quercetin’s antioxidant effects protect cells from the actions of ROS and RNS, which contribute to cancer development. In addition, quercetin contributes to the reduction of ROS-induced injury by increasing the expression of endogenous antioxidant enzymes [145,146].

A second mechanism that may play a role in the anticancer effects of quercetin is the induction of apoptosis by mitochondrial pathways, which involves stimulation of caspase-3 and caspase-9 followed by the release of cytochrome c (Cyt c) and cleavage of poly-ADP-ribose polymerase (PARP) [147,148,149]. Moreover, quercetin also stimulates the proapoptotic genes and inhibits the antiapoptotic genes prompting cell death through the intrinsic and extrinsic pathways of apoptosis [150]. Additionally, quercetin inhibits PI3K/AKT/mTOR and STAT3 pathways specifically in primary effusion lymphoma (PEL) cells, leading to downregulation in the expression of the pro-survival cellular proteins such as c-FLIP, cyclin D1, and c-Myc. Quercetin also reduces the release of IL-6 and IL-10 cytokines, leading to PEL cell death [151]. In parallel, Wang et al. reported that quercetin induces transcription factor EB-mediated lysosome activation and increases ferritin degradation, thus leading to ferroptosis and Bid-involved apoptosis, especially in p53-independent cancer cells [152]. Furthermore, quercetin may have a role in breast cancer therapy. Anti-proliferative effects of quercetin have been demonstrated in the MCF-7 cell line by various mechanisms, including reducing the phosphorylation of P38MAPK, a hallmark of cell proliferation, and influencing the G1 phase and causing apoptosis by suppressing cyclin D1, P21, and Twist expression [153,154]. In thyroid cancer, quercetin plays a role in upregulating Pro-NAG-1/GDF15 in differentiated thyroid cancer cells leading to apoptosis induction and cell cycle arrest [155].

Moreover, quercetin results in cell cycle arrest by its effect on numerous target proteins such as p21, p53, p27, cyclin D, cyclin B, and cyclin-dependent kinases. By preference, quercetin induces cell cycle arrest at the G2/M phase by inauguration of p21 and p73 and cyclin B inhibition, together at the translation and transcription levels [47]. Moreover, quercetin has been shown to inhibit an important key molecule, VEGF, which plays a significant role in the survival of endothelial cells and can cause tumor angiogenesis [156,157]. Similarly, Sharmila et al. revealed that quercetin supplementation also normalizes the following factors: insulin-like growth factor receptor 1 (IGFIR), AKT, androgen receptor (AR), and cell proliferation proteins, which are known to be increased in cancer. In osteosarcoma, Shenglong Li et al. indicated that quercetin assists in reducing the invasion, adhesion, proliferation, and migration rates of human metastatic osteosarcoma cells by inhibiting parathyroid hormone receptor 1 (PTHR1) activity [158]. In addition, quercetin can alter the depolarization of mitochondria and calcium cytoplasmic concentration [159], has a role in increasing the phosphorylation of c-Jun N-terminal kinase, ERK1/2, P38, and P90RSK proteins, and inhibits the phosphorylation of S6, AKT, and P70S6K proteins [159].

In regard to thyroid cancer, the anticancer activity of quercetin was demonstrated through downregulation of Hsp90 levels and reducing chymotrypsin-like proteasome activity [160,161]. As for ovarian cancer, several in vitro studies demonstrated that quercetin inhibits cancer angiogenesis [162] and causes suppression of cell survival, proliferation, migration, and adhesion of the ovarian cancer cell line PA-1 [163]. However, in colon cancer, quercetin inhibits the cell viability of colon 26 (CT26) and colon 38 (MC38) cell lines, while assisting the expression of epithelial–mesenchymal transition (EMT) markers, such as E-, N-cadherin, β-catenin, and snail. In addition, quercetin also inhibits the migration and invasion abilities of the CT26 cell line through modulating the expression of matrix metalloproteinases (MMPs) and tissue inhibitors of metalloproteinases (TIMPs) [164].

Regarding quercetin safety, clinical investigations have demonstrated that quercetin has no toxicity or negative effects [165]. Quercetin has been found in clinical tests to have anti-inflammation and antitumor properties, as well as to alleviate anemia and decrease blood pressure [165].

### 5.4. EGCG (Epigallocatechin Gallate)

EGCG, also known as epigallocatechin gallate, is a natural flavonoid under the subclass of falvan-3-ols (catechins). It is considered a polyphenolic product with a complex structure, and it contains a flavanol core (flavan-3-ols) structure esterified with gallic acid [166]. EGCG can be found in cocoa-based products, nuts, and some fruits, but green tea (*Camellia sinensis Theaceae)* is still considered as the main source of this product [167]. EGCG is considered as the most abundant and therapeutic active catechin in this plant with a percent of nearly 65% among the other three other catechins (epicatechin (EC), epigallocatechin (EGC), and epicatechin gallate (ECG). Due to its phenolic structure, it is very efficient in radical scavenging, and it exhibits its antioxidant effect through several mechanisms, such as hydrogen atom transfer, electron transfer, and the chelation of catalytic metals [168]. Surprisingly, its antioxidant effect is even greater than vitamin E and vitamin C [169,170]. Moreover, it has a role in cancer, diabetes, obesity, cardiovascular, neurodegenerative, and metabolic diseases [102,171,172]; furthermore, it has antiviral, antibacterial, and anti-inflammatory properties [173].

As an anticancer product, EGCG exhibits its action through multiple mechanisms, such as affecting redox reactions and inducing apoptosis and cell arrest, impairing certain proteins and factors that have a role in cancer development, inhibiting angiogenesis, acting as a metal chelating agent, stabilizing p53 for its antitumor activity, and affecting cell proliferation.

EGCG inhibits redox-sensitive transcription factors, such as nuclear factor-κB (NF-κB) [174] and leads to a reduction of NF-κB DNA binding activity, decreasing in the expressions of pro-inflammatory mediators such as interleukin-1β (IL1β), IL-6, IL-8, and tumor necrosis factor-α (TNF-α), as well as downstream enzymes such as poly [ADPribose] polymerase (PARP), COX-2, and iNOS, which eventually leads to decreased infiltration of inflammatory cells [175,176,177].

In addition to this, EGCG induces the apoptosis and cell arrest of cancer cells through several mechanisms. For example, the activation of the apoptosis-related proteins caspase-3, caspase-9, and PARP-1, particularly in MCF-7 breast cancer cells [178], by induction of the apoptotic molecular signals such as Bax, caspases, and cytochrome c (cyt. c), especially in HuCC-T1 cholangiocarcinoma cells [179]; reduction of protein expression of adenosine triphosphate binding cassette subfamily G member 2 (ABCG2) and Bcl-2, also increasing Bax and caspase-3 expression, especially in human esophageal squamous carcinoma cells [180]; and suppression of epidermal growth factor receptor (EGFR)/Ras/Raf/mitogen-activated protein kinase (MAPK) extracellular signal-regulated kinase (ERK) signaling pathway, especially in thyroid cancer [181], by the inhibition of the activation of the PI3K/Akt serine/threonine kinase 1 signaling pathway in H1299 lung cancer cells [182]. In gastric cancer, Zhu et al. showed that EGCG had improved pathological lesions of the precancerous lesions of gastric cancer (PLGC) and enhanced the effect of apoptosis promotion in PLGC rats, and the apoptotic pathway activated by EGCG may be correlated to the inhibition of the PI3K/Akt/mTOR pathway [183]. In head and neck cancer, Amin et al. had shown that a mixture of EGCG and resveratrol induces synergistic apoptosis that is supported by caspase-3 and Poly ADP-ribose polymerase(PARP) cleavages and inhibition of tumor growth; the mixture also inhibited AKT-mTOR signaling with overexpression of constitutively active AKT protected cells from apoptosis [184].

An EGCG and epicatechin combination had shown a higher population of cells in S and G2/M phases when compared to the control, which resulted in inducing cell cycle arrest in HepG2 cells [185]. Moreover, in MCF-7 breast cancer cells, EGCG stimulates cell cycle arrest at the G2/M phase [178]. Furthermore, EGCG directly impairs the activity of urokinase, which is a protein that is overexpressed in cancer and has a crucial role in metastasis development; therefore, impairing its activity results in extracellular matrix degradation, consequently leading to an inhibition of cancer invasion [186]. EGCG is also effective in the inhibition of MMP-2 and MMP-9, which are responsible for degrading the basement membrane and assisting cell invasion and are commonly overexpressed in cancer [187]. EGCG also enhances the ratio of SKOV-3 cells in the G1-phase with an associated decline in the ratio of cells in the S phase and G2-phase of the cell cycle [188]. Additionally, EGCG can inhibit matrix metalloproteinases such as MMP-2 and MMP-9, which are usually elevated in cancer and considered as instruments for degrading the basement membrane and facilitating cell invasion [189]. Furthermore, EGCG inhibits signal transducer and activator of transcription 3 (STAT3), which is an oncogene that supports cell survival, proliferation, motility, and progression of the cancer cells [190]. In addition, EGCG plays an important role in the inhibition of activating protein-1 (AP-1) transcription factor, which is associated with the pathogenesis of cancer [191]. As well, it inhibits the VEGF by modulating the activity of hypoxia-inducible factor 1α (HIF-1α) and NF-kB factor [192]; so, as a result, it inhibits the cancer angiogenesis.

Besides this, EGCG also has a metal chelating feature, and given that certain receptor kinases rely on divalent cations for their action, EGCG can inhibit this reaction by chelating to these cations [193].

Additionally, Zhao et al. reported a direct interaction between EGCG and the tumor suppressor p53 with the disordered N-terminal domain (NTD). The EGCG-p53 interaction disrupts p53 interaction with its regulatory E3 ligase MDM2 and inhibits ubiquitination of p53 by MDM2 in an in vitro ubiquitination assay, likely stabilizing p53 for antitumor activity [194].

Related to breast cancer, Tanaka et al. revealed that EGCG had reduced rRNA transcription and cell proliferation through the activation of KDM2A in MCF-7 cells, and EGCG helps in KDM2A activation by the activation of both 5’ AMP-activated protein kinase (AMPK) and ROS production. Essentially, the inhibition of rRNA transcription and cell proliferation by EGCG was particularly detected in MCF-7 cells; however, it was not detected in non-tumorigenic MCF10A cells [195].

### 5.5. Allicin

Allicin, or diallyl thiosulfinate, is a sulfur-containing volatile oil, and it can be produced by tissue damage of the non-proteinogenic amino acid S-allyl cysteine sulfoxide (alliin), which is catalyzed by the alliinase enzyme [196]. It is extracted from garlic (*Allium sativum* L.) and other Allium species such as onions (*Allium cepa* L.) and shallots (*Allium ascalonicum* L.). Allicin cannot be found while garlic is intact; thus, it is activated by chopping or cutting the garlic cloves [197]. Allicin’s presence is easily noticed because of its distinctive odor [198]. As well, allicin may act as an irritant that increases pain-sensing neurons, and self-medication has resulted in a rash of cases of self-harm [199]. Thus, while tiny levels are appreciated in culinary contexts, excessive allicin consumption is definitely hazardous [199].

Allicin exhibits potential as an anticancer, antifungal, and antibacterial product [197], and it also shows activity in cardiovascular diseases (CVD), as it induces vasodilation, inhibits platelet aggregation, prevents hyperlipidemia, and suppresses cardiac hypertrophy [200].

As an anticancer product, allicin induces its effects due to many mechanisms, such as inducing apoptosis and suppressing the migration and invasion of cancerous cells, having synergistic effects with cancer treatments.

According to its anticancer potential, allicin initiates autophagy-dependent cell killing by suppressing the Akt/mTOR signaling pathway [201]. Moreover, it induces apoptosis through the induction of the caspase-3 pathway [202], and Hussain et al. had indicated that apoptosis induction resulted in cycle cell arrest in the G2 /M phase [203]. Maitisha et al. had shown that allicin induces apoptosis and regulates biomarker expression in breast cancer in vitro due to modulation of the p53 signaling pathway [204]. Moreover, it was found that allicin is capable of inhibiting the expression of VCAM-1 in MCF-7 cells [205]. In addition, allicin can also inhibit telomerase activity and induce the apoptosis of gastric cancer SGC-7901 cells [206]. In glioma, Li et al. had shown that allicin had successfully inhibited proliferation and stimulated apoptosis in U251 glioma cells in vitro and also enhanced the activation of both intrinsic and extrinsic apoptosis signaling pathways in U251 cells [84]. Moreover, Cha had proved that allicin inhibits the cell viability of U87MG cells and stimulates cell death through apoptosis, which is mediated through the Bcl-2/Bax mitochondrial pathway, MAPK/ERK signaling pathway, and antioxidant enzyme systems [207]. In ovarian cancer, allicin can activate the JNK pathway, which leads to mitochondrial Bax translocation and mitochondrial release of cytochrome, thus inducing SKOV3 cell apoptosis in glioblastoma cancer cells.

In breast cancer cell lines, allicin induces cell cycle arrest through targeting the p53 pathway [204]. More and more, allicin suppresses the cell growth of human glioblastoma via inducing G2/M and S phase cell cycle arrest [208]. In cervical cancer, allicin suppresses the migration and invasion in cervical cancer cells mainly by inhibiting NRF2 [209]. In lung carcinoma, it was noticed that it prevents the invasion of lung adenocarcinoma cells by altering TIMP/MMP balance and by lowering the activity of the PI3K/AKT signaling pathway [62].

In gastric cancer, a combination of allicin with 5-Florouricil could decrease multidrug resistance in these cells by lowering the expression of WNT5A, Dickkopf-1(DKK1), multidrug resistance protein 1 (MDR1), P-glycoprotein 1 (P-GP), and CD44 gene levels [210]. Regarding osteosarcoma, Jiang indicated that a combination of artesunate and allicin was proved to have a synergistic effect on osteosarcoma cell proliferation and apoptosis [211]. In melanoma, Jobani had revealed that combination therapy together with allicin and ATRA (all-trans-retinoic acid) significantly decreased the IC50 (half-maximal inhibitory concentration) value obtained for ATRA alone in CD44+ melanoma cells [212]. In hepatocellular carcinoma, it was noticed that a combination of electromagnetic field and allicin can induce apoptosis in vitro and inhibition of proliferation in the HepG2 cell line [213].

### 5.6. Thymoquinone

Thymoquinone (TQ) is a non-toxic major bioactive ingredient obtained from the essential oil of black seed *Nigella sativa* L. It is a monoterpene with the chemical structure (2-Isopropyl-5-methyl-1, 4-benzoquinone), which has been extensively utilized in traditional medicine in the Middle East and Southeast Asian countries owing to its numerous biological actions [214,215]. Thymoquinone is nontoxic and has a wide range of applications in the treatment of many human disorders, including diabetes and cancer [216]. The pharmacological activities of TQ include antioxidant, anti-inflammatory, immunomodulatory, hepatoprotective, antihistaminic, antimicrobial, antidiabetic, anti-epileptic, and chemo-sensitizing, as well as very promising antitumor activity [217,218,219].

TQ anticancer studies have revealed several mechanisms of action, such as regulation of reactive species interfering with DNA structure, modulating various potential targets and their signaling pathways as well as inducing immune system responses in vitro and in vivo. The distinctive anticancer properties of TQ are mainly due to the induction of apoptotic mechanisms, such as activation of caspases, downregulation of precancerous genes, inhibition of the nuclear factor kappa-light-chain-enhancer of activated B cells (NF-κB), antitumor cell proliferation, hypoxia, anti-metastasis, and reduction of side effects when using traditional chemotherapeutic drugs [220,221]. Altering of cell cycle progression is an important step in the inhibition of cancer development and progression. TQ induces G1 phase cell cycle arrest in human breast cancer, colon cancer, and osteosarcoma cells through inhibiting the activation of cyclin E or cyclin D and upregulating p27 and p21, a cyclin-dependent kinase (Cdk) inhibitor [222]. TQ conjugated with fatty acid has potential activity on cell proliferation, apoptosis, and signaling pathways [217]. Conjugation is performed to increase TQ’s capacity to penetrate cell membranes. Several conjugated forms were studied in HCT116 and HCT116 p53−/− colon cancer and HepG2 hepatoma cells in vitro. Treatment with TQ-4-α-linolenoylhydrazone or TQ-4-palmitoylhydrazone was effective in p53-competent HCT116 cells, mediated by an upregulation of p21cip1/waf1 and a downregulation of cyclin E, and associated with an S/G2 arrest of the cell cycle. HCT116 p53−/− and HepG2 cells showed only a minor response to TQ-4-α-linolenoylhydrazone [223]. TQ induces G0/G1 cell cycle arrest, increases the expression of p16, decreases the expression of cyclin D1 protein in DMBA-initiated TPA-promoted skin tumors in mice, inactivates CHEK1, and contributes to apoptosis in colorectal cancer cells [224,225], as well as inducing G1 phase cell cycle arrest in human breast cancer, colon cancer, and osteosarcoma cells through inhibiting the activation of cyclin E or cyclin D and upregulating p27 and p21, a cyclin-dependent kinase (Cdk) inhibitor [222]. Moreover, TQ causes cell arrest at different stages according to the concentration used (25 and 50 µM) in vivo in human mammary breast cancer epithelial cell lines, MCF-7 [226]. In esophageal cancer, TQ encourages G2/M phase cell cycle arrest by increasing the levels of p21 and p53 while remarkably decreasing cyclin A, cyclin B1, and cyclin E expression [227], as well as inducing DNA damage, apoptosis, increased iROS, and cytotoxicity in C6 glioma cells [228]. TQ reduces the elevated levels of serum TNF-α, IL-6, and iNOS enzyme production and enhances histopathological results in Wistar rats with methotrexate-induced injury to the hepatorenal system [229]. Additionally, TQ has a role in reducing the NO levels by downregulation of the expression of iNos, reducing Cox-2 expression and consequently generating PGE2 and reducing PDA cell synthesis of Cox-2 and MCP1 [230,231].

TQ has an effective role in the reduction of endothelial cell migration, tube formation, and suppression of tumor angiogenesis. TQ noticeably reduces the phosphorylation of EGFR at tyrosine-1173 residues and JAK2 in vitro in HCT 116 human colon cancer cells and downregulates the Jak2/STAT3 signaling pathway in human melanoma cells and HL60 leukemia cells [232,233,234]. TQ causes G2/M cell cycle arrest and stirred apoptosis, and it significantly lowers the nuclear expression of NF-κB. Moreover, TQ has a role in the elevation of PPAR-γ activity and downregulation of the gene’s expression for Bcl-2, Bcl-xL, and survivin. Furthermore, it has an antiproliferative effect, especially when combining it with doxorubicin and 5-fluorouracil, which results in increased cytotoxicity in the breast cancer xenograft mouse model [235]. Moreover, TQ has a role in the downregulation of the expression of STAT3-regulated gene products in gastric cancer in both in vivo and in vitro models [236]. Reports showed that TQ plays an essential role in the induction of apoptosis by decreasing the expression of antiapoptotic proteins, as well; it also significantly increased the expression of pro-apoptotic protein [237]. This process is mediated by the activation of caspases 8, 9, and 7 in a dose-dependent manner and increases the activity of PPAR-γ [238,239,240].

TQ prevents DNA damage caused by free radicals by scavenging the free radical activity [241,242,243]. TQ shows a significant effect in the decrease of expressions of CYP3A2 and CYP2C 11 enzymes [244]. TQ treatment showed activity in the reduction of CYp1A2, CYP 3A4, and CYp3A4 enzyme activity and the increase of the phase II enzyme GST. TQ has proven its role in tumor prevention through activation of antioxidant enzymes and its antioxidant activity [245]. TQ treatment illustrates that is has a valuable role in the increase of the PTEN mRNA. Moreover, it has a pivotal role in the inhibition of breast cancer cell proliferation and induction of apoptosis via activation of the P53 pathway in the MCF-7 cell line, the finding having revealed that a time-dependent increase of PTEN occurs in cells treated with TQ as compared with untreated cells [246]. TQ induces degradation of the tubulin subunit in the cells; it also inhibits the telomerase enzyme activity. Furthermore, it causes the suppression of androgen receptor expression and E2F-1, which is essential for the proliferation and viability of androgen-sensitive and androgen-independent prostate cancer cells [217].

Revolutionary findings have revealed TQ’s ability to regulate microRNA (miRNA) expression. MiRNAs are small noncoding RNAs that modulate gene expression and cellular signaling pathways through variation in the features of mRNA, thus moderating TQ’s anticancer effect through P53, PCNA, cyclin D1, Bcl-2, NF-κB, TWIST (Twist1,2), ZEB, eEF-2 K, PI3K/Akt, and Src/Fak signaling pathways [247]. Along with TQ’s impairment of autophagic flux and inhibition of the EMT and cell invasion via activation of the miR-877-5p/PD-L1 (programmed death ligand 1) axis in bladder carcinoma cells and the expression of Beclin-1 and LC3 in triple-negative breast cancer (TNBC) cells, it suppresses the pathways related to cell migration/invasion and angiogenesis, including Integrin-β1, VEGF, MMP-2, and MMP-9 [248,249]. Thymoquinone upregulates miR-125a-5p, attenuates STAT3 activation, and potentiates doxorubicin antitumor activity in murine solid Ehrlich carcinoma [250].

### 5.7. Emodin

Emodin is a Chinese herb-derived anthraquinone isolated from the roots and rhizomes of several plants such as *Rheum palmatum*, *Polygonum cuspidatum, Polygonum multiflorum, Aloe vera*, and *Cassia obtusifolia*, as well as different fungal species, including *Aspergillus ochraceus*, and *Aspergillus wentii*. Its chemical structure is (1,3,8-trihydroxy-6-methylanthraquinone) [251,252,253]. Emodin displays a variety of pharmacological activities, such as antiviral, antibacterial, anti-allergic, anti-osteoporotic, anti-diabetic, immunosuppressive, neuroprotective, hepatoprotective, anti-cardiovascular disease, antitumor, and anti-inflammatory activities [253,254,255].

According to Hsu and Chung’s review (2012), the molecular mechanisms of emodin comprise cell cycle arrest, apoptosis, and the promotion of the expression of hypoxia-inducible factor 1α, GST, *P*,*N*-acetyltransferase, and glutathione phase I and II detoxification enzymes while inhibiting angiogenesis, invasion, migration, chemical-induced carcinogen–DNA adduct formation, HER2/neu, CKII kinase, and p34cdc2 kinase in human cancer cells [256]. Emodin inhibits non-small lung cancer cells via modulation of the cell cycle by decreasing cells in the G2/M and S phases and increasing cells in the G1/G0 phase [257]. Moreover, emodin suppresses hepatocellular carcinoma by inducing G2/M and S phase arrest [258]. In glioma U251 cells, emodin resulted in the induction of cell cycle arrest. Zhou et al. found that the ratio of U251 cells was decreased in the G0/G1 phase compared to the control, and the ratio of cells in the G2/M and S phases was increased when treated with emodin [259]. It has been reported to inhibit tumor-associated angiogenesis through the inhibition of ERK phosphorylation and to suppress VEGFA transcription and thus tumor angiogenesis in triple-negative breast cancer (TNBC), in addition to its antiproliferative and antimetastatic effects [260,261]. Emodin also inhibits Aurora kinase A (AURKA), which plays an essential role in proliferation and is involved in cisplatin resistance in various cancer cells [262]. It downregulates the expression of survivin and β-catenin, inducing DNA damage and inhibiting the expression of DNA repair [256]. It also inhibits the activity of casein kinase II (CKII) by competing at ATP-binding sites [256,263]. Emodin induces necroptosis through ROS-mediated activation of the JNK signaling pathway as well as inhibits glycolysis by downregulation of GLUT1 through ROS-mediated inactivation of the PI3K/AKT signaling pathway [264]. According to some findings, it upregulates hypoxia inducible factor HIF-1 and intracellular SOD and boosts the efficacy of cytotoxic drugs [265,266]. Emodin may sensitize tumor cells to radiation therapy and chemotherapy and inhibit the pathways that lead to treatment resistance. It was found to reverse gemcitabine resistance in vitro in pancreatic cancer cell lines by decreasing the expression of MDR-1 (*P*-gp), NF-κB, and Bcl-2, increasing the expression levels of Bax, cytochrome-C, and caspase-9 and -3, and promoting cell apoptosis unstimulated and in gemcitabine-induced-resistance pancreatic cancer cell lines [267]. Furthermore, in vitro and in vivo findings have concluded that emodin downregulates both XIAP and NF-κB and enhances apoptosis in mice bearing human pancreatic cancer cells [268]. Chemosensitization was also observed in gallbladder cancer, where an independent combination treatment of emodin with cisplatin, carboplatin, or oxaliplatin augmented chemosensitivity in vitro in SGC996 gallbladder cancer cells and in vivo in gallbladder-tumor-bearing mice. Wang et al., 2010, credited these findings to the reduced glutathione level, downregulation of multidrug resistance-related protein 1 (MRP1), and to the increased apoptosis caused by such combinations [269]. Additionally, enhanced chemosensitivity was observed in vitro in DU-145 cancer cell lines (multidrug resistant prostate carcinoma cell line) and in vivo in tumor-bearing mice when treated with a combination of emodin and cisplatin. The mechanism was shown to involve ROS-mediated suppression of multidrug resistance and hypoxia inducible factor-1 in over-activated HIF-1 cells [270].

### 5.8. Genistein

Genistein [4,5,7-trihydroxyisoflavone or 5,7-dihydroxy-3-(4-hydroxyphenyl) chromen-4-one] is an isoflavonoid with a 15-carbon skeleton and is classified as a phytoestrogen. It is found in food (especially legumes) in the glycosylated or free form. It is structurally similar to 17β-estradiol, which is the reason for its ability to bind to and modulate the activity of estrogen receptors [271]. It was isolated for the first time in the year 1899 from *Genista tinctoria*; hence, it was named after the genus of this plant. However, it is the main secondary metabolite of the *Trifolium* species and in *Glycine max* (soybean). In fact, soybean, soy-based foods, and soy-based drinks are the best sources of genistein. Lupin (*Lupinus perennis*) is also a legume that holds similar nutritional value to that of soybean in terms of genistein content. Other important legumes are broad beans and chickpeas, which are known to contain significant amounts of genistein, although less than soybean and lupine. Genistein pharmacologically acts as an anticancer, estrogenic, and anti-osteoporotic agent [271,272].

Genistein exerts its anticancer effects by inducing apoptosis, decreasing proliferation, and inhibiting angiogenesis, as well as metastasis, which was illustrated by decreased tumor growth and development in the hepatocellular cancer models of nude mice [273] and Wistar rats [274], as well as in the gastric cancer model of Wistar rats [275]. Genistein’s role in prostate cancer was extensively studied in vivo in different animal models, such as the Lobund-Wistar rat (which is a unique rat model that spontaneously develops metastatic prostate cancer in 30% of its population), and in SCID mice transplanted with human prostate carcinoma cells (LNCaP, PC3, and DU-145). Some in vivo studies included normal rats to test for genistein’s toxic effect on the prostate and its effect on the expression of the androgen and estrogen receptor [276,277,278].

Genistein is involved in regulation of key biological processes including those in different types of cancer via epigenetic modulation in a direct or indirect manner through estrogen-receptor-dependent pathways, where it was reported to target estrogen receptor (ER), human epidermal growth factor receptor-2 (HER2), and breast cancer gene-1 (BRCA-1) in multiple BC cell lines [279,280]. Genistein was found to inhibit histone deacetylase (HDAC) enzymes, which are responsible of regulating histone acetylation of DNA [272], in MCF-7 and MDA-MB-468, and in immortalized but noncancer fibrocystic MCF10A breast cells at very low, dietary-relevant concentrations [281]. A particular HDAC enzyme is HDAC-6, which is known to acetylate and activate heat shock protein (Hsp90). Basak et al. 2008, reported that the increased ubiquitination of androgen receptors was due to the inhibition of Hsp90 chaperones in genistein-treated LNCaP prostate cancer cells. These results strongly support the hypothesis that genistein may be an effective chemopreventive agent for prostate cancer [282].

It inhibits cyclooxygenase-2 (COX-2) directly and indirectly by suppressing COX-2 stimulating factors such as activated protein-1 (AP-1) and Nf-κB. COX-2 overexpression has been described in pancreatic, colon, breast, and lung cancer, and its inhibition has been correlated with decreased development of cancerous tumors in the esophagus and in the colon [272]. Genistein inhibits CDK by upregulating p21, and it suppresses cyclin D1, which ultimately induces G2/M cell cycle arrest and decreases tumor cell progressions [272,277,283,284,285]. Moreover, in pancreatic cancer cells, it activates G0/G1 phase cell cycle arrest [286]. Genistein was reported to downregulate the expression levels of matrix metalloproteinase-2 (MMP-2) in glioblastoma, melanoma, and breast cancer, as well as regulate caspase-3 and p38MAPK pathways in prostate cancer cell lines. Matrix metalloproteinase (MMP) is the starting step in metastasis and angiogenesis cascade [272,287,288]. In addition, AP-1 is an angiogenic cytokine, which is inhibited by genistein; consequently, such an inhibitory effect will impede several targets, including cyclin D1, MMP, VEGF, Bcl-2, uPA, and Bcl-XL [272]. Genistein induces *Cd74* downregulation, which regulates the NF-κB/Bcl-xL/TAp63 signaling pathway by contributing to its therapeutic effect on triple-negative breast cancer (TNBC) tumors. Additionally, genistein can modify expression levels of key epigenetic-associated genes such as DNA methyltransferases (*Dnmt3b*), ten-eleven translocation (*Tet3*) methylcytosine dioxygenases and histone deacetyltransferase (*Hdac2*), and their enzymatic activities, as well as genomic DNA methylation and histone methylation (H3K9) levels [289].

Moreover, genistein can influence metastasis and induce apoptosis by inhibiting Akt, as well as NF-κB cascades, in PC3 cell lines and MDA-MB-231 breast cancer cell lines, as well as inhibiting the IL-6/STAT3 pathway in MDA-MB-453 breast cancer cell lines, thus inhibiting cell proliferation [272,290,291]. Furthermore, genistein decreases phosphorylated-Akt in HT-29 colon cancer cells [292], in LNCaP prostate cancer cells [293], and in HeLa and CaSki cervical cancer cell lines [294], as well as in other cancer cell cultures [272].

This concluded that genistein modulates its anticancer activity via several signaling pathways both at transcription and translation levels through regulation of several key cellular pathways [295]. Moreover, Lui et al. 2021, successfully demonstrated that genistein can specifically bind to DNA-dependent protein kinase catalytic subunit (DNA-PKcs) and block the DNA-PKcs/Akt2/Rac1 pathway, thereby effectively inhibiting radiation-induced invasion and migration of glioblastoma multiforme (GBM) cells in vitro and in vivo [296]. In addition, this phytochemical synergistically reverses the resistance mechanism of standard chemotherapeutic drugs, increasing their efficacy against BC [297].

### 5.9. Parthenolide

Parthenolide (PTL) is a natural product that belongs to the class of germacrane sesquiterpene lactones, and it is the most outstanding representative of its subclass of germacranolides [298]. The PTL is produced as a secondary metabolite by plants of the Asteraceae/Compositae (daisies) and Magnoliaceae family (magnolias) [299]. It is extracted from leaves of the medicinal plant feverfew (*Tanacetum parthenium*) [299,300]. Basically, PTL contains an α-methylene-γ-lactone ring and epoxide group, which contribute to its ability to interact with nucleophilic sites of biological molecules [298,299,300]. It displays redox-modulating, epigenetic, anti-inflammatory, and anti-bacterial biological activities [301]; thus, its use in the treatment of migraines and arthritis has been confirmed [299,302]. Moreover, it has shown potent anticancer activity against various types of cancer, including colorectal, pancreatic, lung, skin, melanoma, bladder, and breast cancer [303,304]. Parthenolide is not only a powerful anticancer drug, but it also has no appreciable toxicity to normal cells at the cytotoxic effective concentration [305].

One of the several mechanisms underlying the PTL’s antitumor activity is its ability to stimulate apoptosis through inhibition of the nuclear factor kappa B (NF-κB) pathway [306]. This pathway plays a major role in the expression of pro-inflammatory genes, including cytokines, chemokines, and adhesion molecules [307]. Essentially, the NF-κB pathway activates and induces cell survival in primary effusion lymphoma (PEL). PEL is a rare aggressive disorder that is frequently caused by human herpesvirus 8 (HHV-8) infection [308,309]. PTL can induce apoptosis by the initiation of DNA hypomethylation, histone acetylation, enhancement of oxidative stress via endoplasmic reticulum or thiol depletion, and activation of p53. In addition, PTL has shown its ability to significantly inhibit cell growth and induce G0/G1 cell cycle arrest [310] via several mechanisms such as transcriptional regulation, signal transduction modulation, and induction of oxidative stress [78]. In prostate cancer cells, PTL induces cell cycle arrest via the inhibition of miR-375 [311]. Additionally, PTL can inhibit ubiquitin-specific peptidase 7, which is a deubiquitinating enzyme, by direct interaction [9]. Needless to say, ubiquitination is crucial in various cellular biological activities; therefore, dysregulation of this process can lead to cancer development [312]. A study confirmed that PTL inhibits USP47, which regulates colorectal cancer stem cells (CCSCs). Therefore, PTL could effectively suppress CCSCs’ renewal and stemness maintenance, providing a potential therapy for colorectal cancer [313].

Another study showed that PTL significantly inhibited cell proliferation and migration in two lung cancer cell lines (A549 and H1299). In terms of the involved mechanism, PTL has the capability of blocking the phosphorylation of insulin-like growth factor 1 receptor, Akt, and forkhead box O3α [303]. Further, it was shown that PTL has powerful cytotoxicity towards human non-small cell lung cancer cells via targeting the B-Raf and inhibiting the MAPK/Erk signaling pathway [314]. Hence, PTL can be considered a novel therapeutic strategy for renal cell carcinoma. A study conducted by Li et al. revealed the ability of PTL to inhibit the oncogenic characteristics of 786-O and ACHN cells, decrease the viability of cancer cells, and suppress the formation of mammospheres [315]. PTL induces autophagy by inhibiting the PI3K/AKT/mTOR signaling pathway through the activation of phosphatase and tensin homolog expression [316]. Additionally, it can covalently modify the cysteine 427 of focal adhesion kinase 1 (FAK1), resulting in damaging FAK1-dependent signaling pathways [317]. Another inducing autophagy mechanism of PTL was confirmed in pancreatic cancer via the formation of autophagosomes and conversion of LC3A to LC3B form [318].

In addition, PTL treatment converses the epithelial to mesenchymal transition (EMT) process by significantly decreasing the mesenchymal marker vimentin and inducing the epithelial marker E-cadherin protein expression in both MCF-7 and MDA-MB-231 cells [319]. The reversal of EMT was concomitant with the reduction of the TGFβ protein, gene expression [320], and the EMT-inducing transcription factor TWIST1 gene expression. Therefore, it prevents cancer progression and metastasis [319,320]. Additionally, PTL decreased the viability of C918 and SP6.5 cells, which are human uveal melanoma cells [321].

Moreover, PTL is a strong inhibitor of Janus kinases (JAKs) [304]. PTL covalently alters the Cys243, Cys178, Cys335, and Cys480 of JAK2, leading to inhibition of their kinase activity and blocking the signal transducer and activator of transcription 3 (STAT3) signaling pathway [322].

In cervical cancer, PTL showed a potent anticancer activity through inhibiting HeLa cell viability in a dose-dependent manner and inducing the generation of ROS that result in loss of mitochondrial membrane potential [323].

### 5.10. Luteolin

Luteolin is a flavone compound that belongs to the flavonoids group [324]. Generally, it can be found in flowers (*Reseda luteola and Chrysanthemums*), herbs (parsley, peppermint, oregano, and thyme), vegetables (celery seeds, onion leaves, cabbages, sweet bell peppers, carrots, and broccoli), and spices (cardamom and anise) [324,325]. Chemically, it is a 3,4,5,7-tetrahydroxyflavone that has a C6-C3-C6 structure with two benzene rings and one oxygen-containing ring with a C2-C3 carbon double bond [326]. It has been proposed that luteolin has multiple cardio-protective [327], as well as anti-microbial [328], anti-inflammatory, anti-oxidant [329,330], and anticancer effects [331]. Moreover, luteolin displays anticancer activity against colon, liver, lung, skin, and breast cancer [332,333]. On the other hand, luteolin showed a wide range of safety toward normal cells. For example, when luteolin was delivered at a dose of 100 mg/day in human clinical studies, there was no dose-limiting harm (17).

Luteolin exhibits its anticancer properties through different mechanisms, including blocking the activity of epigenetic targets such as DNA methyltransferases, some classical histone deacetylases, and SIRT1 [331]. It also induces autophagy and cell apoptosis and inhibits invasion, network formation, and migration [334]. For example, this compound has the ability to inhibit U-251 cell migration and tumorigenesis [335]. Correspondingly, it causes apoptosis in tamoxifen-resistant breast cancer via activation of apoptosis-related proteins (cleaved poly (ADP-ribose) polymerase, cleaved-Caspase-7, 8, 9) [336]. Moreover, luteolin upregulates microRNA-6809-5p and inhibits hepatocellular carcinoma’s (HCC) cell growth [337]. Overexpression of miR-6809-5p suppresses the expression of flotillin 1 [338] and inactivates signaling pathways, including Erk1/2, p38, JNK, and NF-κB/p65 in HCC cells.

Consequently, it suppresses cancer development [337]. Likewise, luteolin has a dual action in downregulating/upregulating autophagy in cancer therapy [41]. For instance, in vitro study has shown that luteolin has an antitumor effect in the HCC cell line SMMC-7721 by inducing autophagy [339]. Similarly, in mice with liver cancer, luteolin has improved the host’s system via various mechanisms such as modifying the levels of α-fetoprotein and marker enzymes, as well as reducing the levels of glutathione and the inflammatory cytokines interleukin-2 and interferon-γ [340].

Moreover, luteolin increases the expression of miR-124-3p and activates the death receptor and mitogen-activated protein kinase (MAPK) signaling pathways in glioma [341]. Further, luteolin raises the level of intracellular ROS [325]. This can be achieved by activation of the lethal endoplasmic reticulum stress response, mitochondrial dysfunction in glioblastoma cells, as well as activation of ER stress-associated protein expressions [342]. In metastatic human colon cancer, luteolin reduces the viability and proliferation of SW620 cells and increases the expression of antioxidant enzymes. In addition, the expression of antiapoptotic protein Bcl-2 decreases, whereas the expression of proapoptotic proteins Bax and caspase-3 increases [343]. Additionally, this phytochemical inhibits the growth of HCT116 colon cancer cells via p53-dependent regulation of cell cycle arrest [344]. Additionally, in MDA-MB-231 breast cancer cells, luteolin has shown its ability to induce cell cycle arrest in the S phase in a dose-dependent manner [345]. Moreover, a study revealed that a combination of luteolin and baicalein (a 5,6,7-trihydroxyflavone) inhibits the growth of colon cancer cells of the drug-sensitive LoVo cell line and its drug-resistant LoVo/Dx subline as well [333].

Interestingly, luteolin and quercetin synergistically enhance the anticancer effect of 5-fluorouracil (5-FU) in HT 29 cells and result in minimizing the toxic effects of 5-FU in the treatment of colorectal cancer [44]. Luteolin can remarkably induce DNA double-strand breaks (DSBs) in DT40 cells and induce sensitivity and defects in DSB repair in Ku70 cells. Furthermore, it improves the formation of Top2cc in Ku70 cells [346]. Luteolin and its derivative apigenin significantly inhibit lung cancer cell growth and downregulate the IFN-γ-induced PD-L1 expression by suppressing the phosphorylation of STAT3 [347]. Luteolin has demonstrated its ability to suppress the expression of TAM receptor tyrosine kinases and the activation of Axl, which in turn leads to the inhibition of cell proliferation in both parental and non-small cell lung cancer cells [348]. Recent studies have shown that luteolin regulates cyclin D1, cyclin E, p21, and Bcl2, resulting in the prevention of cancer development in gastric cancer. In addition, luteolin controls metastasis by regulating MMPs expression and the EMT process [349].

Interestingly, Namkung et al. reported that luteolin potently inhibits Anoctamin 1 (ANO1) chloride channel activity and decreases its protein stability in prostate cancer [350]. It is acknowledged that overexpression of ANO1 is involved in the tumorigenesis of epithelial cancers [351]. Furthermore, luteolin significantly reduces ribosomal protein S19 expression by blocking the Akt/mTOR/c-Myc signaling pathway in cancer cells [352]. Table 1 summarizes the mentioned natural products with their main natural sources and anticancer mechanisms of action. Figure 2 demonstrates the effects of natural products on cancer hallmarks.

## 6. Clinical Studies

### 6.1. Curcumin

Meriva (curcumin phytosomes) was investigated in controlled semi-quantitative clinical research to determine its potential to relieve the side effects of cancer radiotherapy and chemotherapy. This formulation was also studied for six weeks as an adjuvant to chemotherapy in a group of solid tumor patients (at 1500 mg/day in three separate doses). Patients’ quality of life was greatly improved, and systemic inflammation was significantly reduced [354].

Curcumin’s safety and anticancer effectiveness in human colon cancer patients were proven in clinical research conducted by Shehzad and colleagues [355]. Curcumin was proven to accumulate at the colorectum and acquire the effective therapeutic concentration in a phase I clinical trial in which curcuma extract was provided orally to patients with colorectal cancer at dosages of up to 2.2 g daily (equal to 180 mg of curcumin) for several months [356].

There were improvements in all categories of the International Prostate Symptom Score in a pilot study to test the efficacy of Meriva in benign prostatic hyperplasia (BPH) (given 1000 mg/day in two divided doses) [357].

Docetaxel plus curcumin administered to individuals with advanced breast cancer with dose escalation demonstrated some improvements in biological and clinical responses in the majority of patients in a phase I trial [358]. Another study looked into the efficacy of combining curcumin and quercetin to regress adenomas in patients with familial adenomatous polyposis (FAP), an autosomal dominant condition characterized by colon and rectum cancer. For six months, FAP patients with prior colectomy were administered 480 mg curcumin and 20 mg quercetin orally three times daily. The quantity and size of these malignant polyps were dramatically reduced [359].

In a phase II experiment, twenty-five patients with advanced pancreatic cancer were given 8 g of curcumin capsules daily, with restaging every eight weeks [360]. There was no toxicity seen when the medication levels peaked at 22 to 41 ng/mL. This study found that, despite its low absorption, oral curcumin had biological action and was safe in some pancreatic cancer patients. Furthermore, the expressions of NF-κB, COX-2, phosphorylated signal transducer, and activator of transcription 3 (which were higher in patients compared to healthy volunteers) were found to be lowered in most patients’ peripheral blood mononuclear cells [360].

A randomized double-blind placebo-controlled parallel-group clinical study was conducted in a group of 150 women with advanced breast cancer to examine the efficacy and safety of an intravenous infusion of curcumin in conjunction with paclitaxel. Three months were spent monitoring the patients. This study concluded that using curcumin in conjunction with paclitaxel was preferable to using paclitaxel alone [361].

According to the research, derivatizing curcumin may be a viable option for increasing target selectivity and producing anticancer therapeutic candidates with greater potency [362]. Twenty patients with cancer anorexia–cachexia syndrome who were properly nourished via a feeding tube were included and randomly assigned in a 1:1 ratio to receive oral curcumin (at a dose of 4000 mg daily) (*n* = 10) or placebo (*n* = 10) for 8 weeks [363]. Curcumin has a statistically significant advantage on improving muscle mass when compared to the placebo. The addition of curcumin resulted in a considerable increase in muscle mass compared to the usual nutritional assistance. Furthermore, it may improve and postpone the decline of other body composition metrics, such as handgrip strength and absolute lymphocyte count [363].

### 6.2. Resveratrol

Resveratrol has been studied in clinical trials for its potential therapeutic efficacy in a variety of disorders such as cancer, obesity, neurological disorders, cardiovascular disorders, and infections. This review focuses on the role of resveratrol as an anticancer agent [364].

The safety of resveratrol has been tested in healthy persons, and it has been shown to be safe up to doses of 5 g/d [365]. The most prevalent tumors that have been shown to respond positively to resveratrol include colon cancer, breast cancer, and multiple myeloma [365].

The effects of resveratrol on the expression of several cancer-related genes, such as CCND-2, p16, RASSF-1α, and cancer-promoting prostaglandin E2 (PGE2), were investigated in a randomized placebo-controlled clinical research in women with a high risk of breast cancer [366]. For 12 weeks, the participants were given two capsules per day containing either placebo, 5 mg of trans-resveratrol, or 50 mg of trans-resveratrol. PGE2 levels were discovered to be reduced [366].

A phase I trial was carried out to evaluate the effect of low-dose resveratrol (80 mg/d) and resveratrol-containing freeze-dried grape powder (GP) (80 g/day, equivalent to 450 g fresh grapes) on colon cancer. After 14 days of treatment, there was an increase in the expression of Myc and cyclin D1 in colon cancer tissue. When compared to resveratrol, the GP had more dramatic effects [367]. Furthermore, in normal colonic mucosa, GP strongly lowered CD133 with a modest effect on LGR5 (downregulation of CD133 and LGR5 is associated with growth inhibition of colon cancer cells) [368].

Micronized resveratrol (SRT501) was employed in another investigation to improve resveratrol absorption across the gastrointestinal system. SRT501 was given to individuals with colorectal cancer and hepatic metastases at a dose of 5 g per day for two weeks. SRT501 was well tolerated and increased mean plasma resveratrol levels (3.6-fold) after a single dosage compared to non-micronized resveratrol [369].

Colorectal cancer is routinely treated with 5-fuorouracil, an efficient chemotherapy medication [370]. The use of resveratrol can improve 5-fluorouracil-induced cytotoxicity while also mitigating the undesirable side effects. The purpose of this study was to look at the possible therapeutic effects of resveratrol in conjunction with 5-FU against colorectal cancer [370]. Eleven participants participated in the three-year trial. Oral resveratrol administration has been found to help prevent colon cancer in people [306]. In addition, the FDA has approved a clinical trial for gastrointestinal tumors (NCT01476592) [371]. The effect of resveratrol on notch-1 signaling, the effect on patient therapy, and patient toleration of resveratrol administration were all examined throughout this trial. It was discovered that resveratrol injection enhanced cleaved caspase-3 levels in malignant tissue, implying that apoptosis in cancer cells was boosted [371].

### 6.3. Quercetin

A phase I clinical investigation to assess quercetin’s safety profile found that it can be safely delivered as an intravenous bolus injection. Quercetin has also been proven to inhibit lymphocyte tyrosine kinase [372].

There is only one completed clinical trial using quercetin in cancer prevention and treatment (chemotherapy-induced oral mucositis), but no findings have been published (https://clinicaltrials.gov/14J (accessed on 14 July 2022)).

### 6.4. EGCG (Epigallocatechin Gallate)

The only epidemiologic study (a nested case–control study within the Shanghai Cohort Study) that investigated the relation of specific tea catechins to esophageal cancer risk used validated urinary biomarkers for tea polyphenol uptake and metabolism. This study’s findings indicated a lower risk for both esophageal and gastric cancer with the existence of EGC in urine, with a greater inverse association in nonsmokers, non-alcohol drinkers, or those with lower serum levels of carotenes [373].

Similarly, a cohort study on 481,563 volunteers aged from 51–71 years also revealed a statistically significant inverse connection between hot tea drinking and risk of pharyngeal cancer after up to eight years of follow-up [374]. Moreover, a randomized placebo-controlled phase II clinical trial assessed the efficacy of green tea extract (both oral administration and topical treatment) on oral mucosa leukoplakia (precancerous lesion of oral cancer) in 59-patients, recording smaller oral lesions in 37.9% of patients who received green tea treatment [375].

Furthermore, a more recent pooled analysis of six cohort studies similarly revealed a statistically significant, inverse link between green tea drinking and stomach cancer risk in women (particularly among female nonsmokers) but not in males [376].

The relationship between green tea consumption and colorectal cancer risk was assessed in a prospective study (a cohort of 69,710 Chinese women aged 40 to 70 years, the majority of whom were lifelong nonsmokers or non-alcohol drinkers), with two to three years of a follow-up. The study’s results indicated that drinking tea on a regular basis considerably lowered the incidence of colorectal cancer [377].

In another prospective trial, tea catechins have been shown to protect against the development of colon cancer. This study investigated the relationship between urine levels of EGC, 4-*O*-methyl-epigallocatechin (4-MeEGC), and EC and its metabolites, in one hand, and the risk of developing colorectal cancer in the other. Results have shown that individuals with higher urine catechin levels had a decreased risk of developing colon cancer [373].

Moreover, in a pilot clinical research including ten female patients (age 38–55 years) with locally advanced noninflammatory breast cancer and undergoing radiation, patients were treated with oral EGCG capsules (400 mg) three times daily for two to eight weeks. Results of this study have shown that EGCG increased the efficacy of radiotherapy in breast cancer patients, which may be a promising possibility for using EGCG as a therapeutic adjuvant in the treatment of metastatic breast cancer [378].

On the other hand, a meta-analysis of seven epidemiological studies (two cohort, one nested case–control, and four case–control) to assess the influence of green tea consumption on breast cancer risk found an inverse correlation only in case–control data [379].

Regarding the effect of green tea on prostate cancer, in one study, which included 42 patients with androgen-independent prostate cancer, only one patient showed a 50% decrease in prostate-specific antigen (PSA) level, which lasted for two months [380]. In the other study, which looked at the efficacy and toxicity of standardized green tea extract on prostate cancer, delayed disease progression was displayed in 40% of patients who completed the treatment [381].

In regard to liver cancer, the findings of a randomized placebo-controlled phase II clinical research indicated that green tea polyphenols protect against two established liver cancer risk factors, aflatoxin and hepatitis B [382]. Moreover, a population-based case–control research observed a statistically significant inverse correlation between green tea drinking and the risk of pancreatic cancer [383].

In patients with papillomavirus-infected cervical lesions, EGCG delivered as a capsule (200 mg p.o. for 12 weeks) was found to be effective [384]. In some malignancies, EGCG promotes cell death via the intrinsic route and inhibits the EGFR, STAT3, and ERK pathways [384]. EGCG modifies and inhibits ERK1/2, NF-B, and Akt-mediated signaling in tumor cells, affecting the Bcl-2 family protein ratio and activating caspases [384].

Finally, a meta-analysis of 22 research studies established that drinking green tea considerably reduced the incidence of lung cancer. Nevertheless, this reduced risk was limited to nonsmokers, and the association was slightly higher in prospective cohort studies than in retrospective case–control studies [385].

### 6.5. Allicin

A double-blind randomized controlled trial involving individuals with colorectal adenomas found that taking a high dose of aged garlic extract was related with a significantly lower risk of developing new colorectal adenomas [386].

Garlic is aged for up to 20 months in aged garlic extract (AGE), a procedure that changes the odorous, harsh, and irritating components of garlic into stable and nontoxic sulfur compounds [387].

Colorectal cancer incidence and growth have been shown to be reduced by aged garlic extract [388]. In patients with advanced digestive system cancer, administering aged garlic boosted natural killer (NK) cell activity but did not improve quality of life (QoL) [389]. Furthermore, high dosages of allitridum and microdoses of selenium have been found to prevent stomach cancer, particularly in men [390].

A meta-analysis of 19 case–control and two cohort studies found a reduction in the incidence of stomach cancer with an increase of 20 g/day of total Allium vegetables such as garlic, onion, leeks, Chinese chives, scallions, and garlic stalks [391].

Other meta-analysis comprising 14 case–control studies on the effect of Allium vegetables on stomach cancer and five case–control studies on the effect of garlic on stomach cancer indicated that Allium vegetables may have a cancer-preventive effect on stomach cancer [392].

A study comprising 343 patients with esophageal squamous cell carcinoma and 755 cancer-free controls concluded that ingesting raw garlic/onion at least once per week significantly protects against esophageal squamous cell carcinoma [393].

Individuals in the highest of three intake groups of total Allium vegetables had a 53% lower incidence of prostate cancer compared to those in the lowest intake category in a population-based case–control study [394].

### 6.6. Thymoquinone

Despite a great number of in vitro and in vivo studies demonstrating thymoquinone’s potential as an anticancer drug, there is no therapeutic application. There are no thymoquinone clinical trials listed on (https://clinicaltrials.gov/ (accessed on 14 July 2022)).

### 6.7. Emodin

Only one study on the effect of emodin on breast cancer was found on the https://clinicaltrials.gov/ (accessed on 15 July 2022) website when searching for clinical studies on the anticancer activity of emodin (ClinicalTrials.gov Identifier: NCT01287468).

### 6.8. Genistein

Breast cancer incidence was found to be lower in peri-menopausal women who ate more soy food [395]. Another meta-analysis of 9000 breast cancer patients found that increasing the genistein dose reduced breast cancer risk [396]. In vitro, however, low dosages of genistein prevent breast cancer cell proliferation, whereas high amounts enhance it [397].

The level of the prostate cancer biomarker prostate-specific antigen (PSA) in blood was reduced in a randomized placebo-controlled double-blind phase II clinical trial in which prostate cancer patients received genistein before radical prostatectomy compared to the control [398].

Genistein, in combination with gemcitabine and erlotinib, can help kill more tumor cells by making tumor cells more susceptible to the medications, according to a phase II trial (ClinicalTrials.gov Identifier: NCT00376948) on patients with locally advanced or metastatic pancreatic cancer.

A randomized phase II experiment on patients with early-stage prostate cancer (ClinicalTrials.gov Identifier: NCT01325311) found that cholecalciferol (200,000 IU) and genistein (as G-2535, which gives 600 mg of genistein) may reduce cancer cell development and be an effective treatment for prostate cancer.

It was discovered in a phase I/II pilot clinical trial (ClinicalTrials.gov Identifier: NCT01985763) to evaluate the efficacy of genistein in the treatment of metastatic colorectal cancer alone or in combination with 5-fluorouracil and platinum compounds that genistein can suppress Wnt signaling, a pathway induced in the majority of colorectal cancers, and can enhance growth inhibition when combined with 5-fluorouracil and platinum compounds. The most important clinical studies were mentioned in Table 2.

### 6.9. Parthenolide

On the https://clinicaltrials.gov/ (accessed on 15 July 2022) website, there are no clinical studies that have been undertaken to assess parthenolide’s anticancer efficacy.

### 6.10. Luteolin

Only one clinical trial evaluating the efficacy of luteolin (nano-luteolin) on tongue cancer was found on clinicaltrials.gov, and it has passed its completion date with an unknown status.

## 7. Conclusions

Natural products extracted from plants are a rich source for anticancer agents. Multiple cancer hallmarks are targeted by plant-derived natural products through altering diverse signaling pathways. Some natural products such as curcumin and resveratrol exhibit the ability to target cancer through multiple mechanisms. Apoptosis induction is the most common pathway activated by plant-derived natural products. Targeting drug resistance and metastasis inhibition were also reported as anticancer mechanisms of these compounds. The use of plant-derived natural products as an adjuvant therapy with conventional treatments is a productive strategy that deserves further investigations to improve cancer treatment protocols.

## Figures and Tables

**Figure 1 molecules-27-04818-f001:**
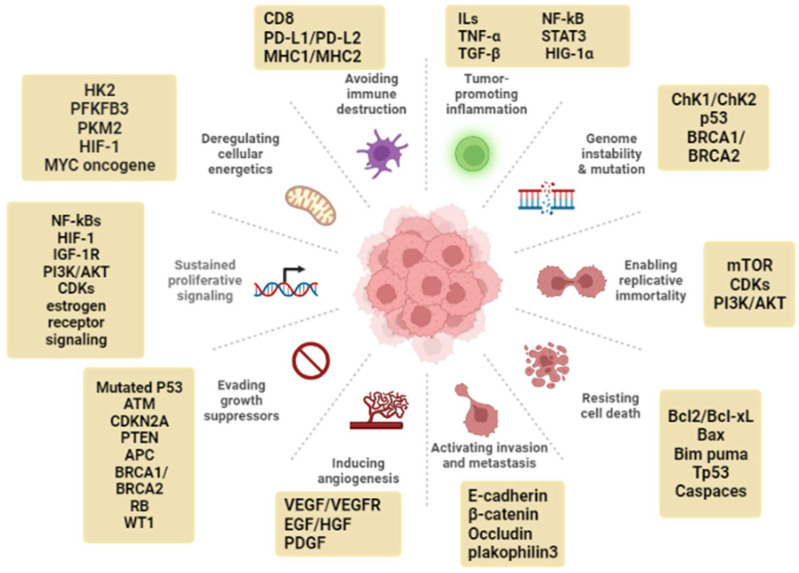
Cancer hallmarks with the main regulatory biomarkers. VEGF, vascular endothelial growth factor; VEGFR, vascular endothelial growth factor receptor; EGF, epidermal growth factor; hepatocyte growth factor; ATM, ataxia-telangiectasia mutated; PTEN, phosphatase and tensin homolog; APC, adenomatous polyposis coli; BRCA1 and BRCA2, breast cancer gene 1 and 2; RB, retinoblastoma; WT1, Wilms tumor; CDKs, cyclin-dependent kinase; IGF-1R, insulin-like growth factor receptor; HIF-1, hypoxia-inducible factor-1; HK2, hexokinase 2; PKM2, pyruvate kinase isoform M2; PFKFB3, 6-phosphofructo-2-kinase/fructose-2,6-biphosphatase 3; PD-L1, programmed death ligand 1; MHC, major histocompatibility complex; CD8, cluster of differentiation 8.

**Figure 2 molecules-27-04818-f002:**
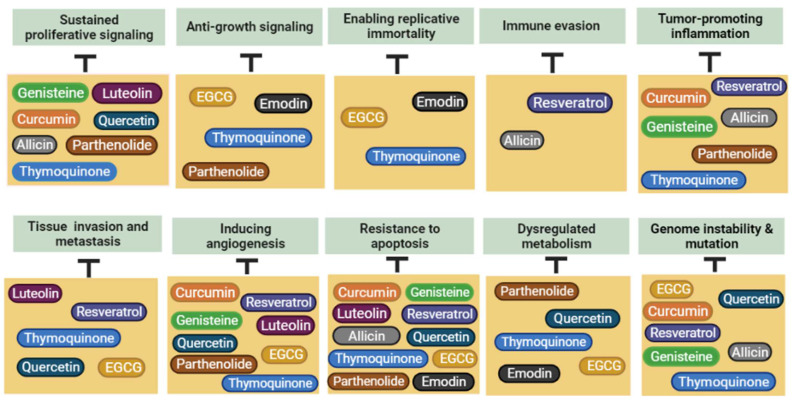
The effects of natural products on cancer hallmarks.

**Table 1 molecules-27-04818-t001:** Plant-derived compounds with their main natural sources and anticancer mechanisms of action.

Compounds	Natural Sources	Mechanisms of Anticancer Activity	References
**Curcumin** 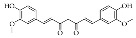	Found in the rhizome of *Curcuma longa* and in others Curcuma multiple species	↓ ROS ↑ Death receptor 5 ↑ caspase-3 and caspase-8 ↓ Bcl-2, NF-kB, EGFR ↑ JNK/ERK/AP1 pathway ↓ PKM2, Lactate production, glucose uptake ↓ miR-181b, miR-203, miR-9, miR-19, miR-21, miR203, miR-9, and miR-208 ↑ activity of chemotherapy (docetaxel, cisplatin, doxorubicin, vincristine) ↓ chemo-drug resistance	[87,88,89,90,92,93,97,102,103,104,105,107]
**Resveratrol** 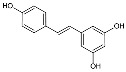	Found in at least 72 plant species, such as mulberries, peanuts, cranberries, blueberries and grapes	↑ SOD, CAT, GS-R-, GPx, GST ↑ chemo-sensitivity of temozolomide, cisplatin, doxorubicin, 5-fluorouracil, gemcitabine, docetaxel, and paclitaxel ↓ EFG, ERK, VEGF ↓ TNF-α, NF-kB, p65 ↑ caspase-3, caspase-9, Bcl-2 associated X protein, p53 ↓ AMPK/mTOR signaling pathway ↓ PI3K/Akt/NF-κB signaling pathway ↓ glioma-associated oncogene homolog 1 ↑ expression of activating receptors on natural killer (NK) cells	[118,120,122,125,126,127,128,130,131]
**Quercetin** 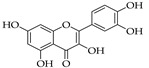	Found in many foods such as capers, apples, berries, Brassica vegetables, grapes, pepper, asparagus, onions, broccoli, shallots, cherries, tea, and tomatoes	↓ ROS ↑ caspase-3, caspase-9 ↓ PI3K/AKT/mTOR and STAT3 pathways ↓ c-FLIP, cyclin D1, and c-Myc ↓ P38MAPK, cyclin D 1, P21 ↓ VEGF, AKT, IGFIR, AR, PTHR1 ↑ c-Jun N-terminal kinase, ERK1/2, P38, and P90RSK ↓ S6, AKT, and P70S6K ↓ Hsp90 levels ↑ E-, N-cadherin, β-catenin, and snail ↓ MMPs	[145,146,147,148,149,151,153,154,156,157,158,159,160,161,164]
**EGCG** 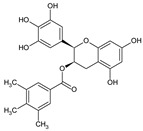	Found in cocoa-based products, nuts, some fruits, but green tea (*Camellia sinensis Theaceae)* is still considered as the main source of this product	↑ caspase-3, caspase-9, PARP-1, Bax ↓ NF-κB, IL1β, IL-6, IL-8, TNF-α ↓ COX-2, iNOS, ↓ ABCG2, Bcl-2 ↓ ERK, EGFR, MAPK ↓ PI3K/Akt/mTOR pathway ↓ MMP-2, MMP-9 ↓ STAT3, AP-1 ↓ VEGF, HIF-1α ↑ P53, KDM2A	[174,175,176,177,178,179,180,181,182,183,184,187,189,190,192,194,195]
**Allicin** 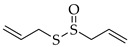	Extracted from garlic (*Allium sativum* L.) and other Allium species such as onion (*Allium cepa* L.) and shallot (*Allium ascalonicum* L.)	↓ Akt/mTOR signaling pathway ↑ caspase-3, p53 ↓ VCAM-1 ↓ NRF2 ↑ Bcl-2/Bax mitochondrial pathway, MAPK/ERK signaling pathway ↓ PI3K/AKT signaling pathway ↓ chemo-drug resistance ↓ MDR1, MDR1, CD44, DKK1, WNT5A	[62,64,201,202,204,205,207,209,210]
**Thymoquinone** 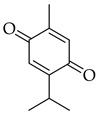	Obtained from the essential oil of black seed *Nigella sativa* L.	↓ NF-κB, cell proliferation, hypoxia ↓ cyclin E, cyclin D ↑ p27 and p21, miR-125a-5p ↑ p21cip1/waf1 ↑ G1 phase cell cycle arrest ↓ TNF-α, IL-6, iNOS, COX-2 ↓ EGFR, JAK2, ↓ Jak2/STAT3 signaling pathway ↓ Bcl-2, Bcl-xL, survivin ↑ chemotherapy sensitivity ↓ STAT3 ↑ caspases 8, 9, 7, PPAR-γ ↓ CYP3A2 and CYP2C 11, CYP 3A4 enzymes ↑ PTEN mRNA, p53 ↓ androgen receptor expression and E2F-1 t ↓ miR-877-5p/PD-L1 ↓ Integrin-β1, VEGF, MMP-2 and MMP-9	[217,220,221,222,223,224,225,228,229,230,231,232,233,234,235,236,238,239,240,244,245,246,248,249,250]
**Emodin** 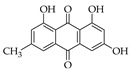	Isolated from the roots and rhizomes of several plants such as *Rheum palmatum*, *Polygonum cuspidatum, Polygonum multiflorum, Aloe vera*, and *Cassia obtusifolia*, as well as different fungal species, including *Aspergillus ochraceus* and *Aspergillus wentii*	↓ HER2/neu, CKII kinase, CKII kinase ↓ ERK, VEGFA ↓ TGF-β ↓ Aurora kinase A ↓ survivin, β-catenin ↓ DNA repair ↓ casein kinase II ↓ GLUT1, PI3K/AKT signaling pathway ↑ HIF-1, intracellular SOD ↓ MDR-1, NF-κB, Bcl-2, XIAP ↑ Bax, cytochrome-C, caspase-9 and -3 ↓ MRP1 ↑ chemo-sensitivity	[256,260,261,262,263,264,265,266,269,178]
**Genistein** 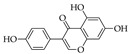	Isolated from *Genista tinctoria* and the main secondary metabolite of the *Trifolium* species and in *Glycine max* (soybean)	↓ histone deacetylase ↓ Hsp90 chaperones ↓ COX-2, NF-kB, AP-1 ↑ p21, caspase-3, p38MAPK ↓ cyclin D1 ↑ G2/M cell cycle arrest ↓ MMP-2, VEGF, Bcl-2, uPA, Bcl-xL ↓ AKT, IL-6/STAT3 pathway ↓ DNA-PKcs/Akt2/Rac1 pathway ↑ chemo-sensitivity	[272,282,290,291,296,297]
**Parthenolide** 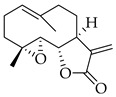	Extracted from leaves of the medicinal plant feverfew (*Tanacetum parthenium*)	↓ NF-Kb ↑ G0/G1 cell cycle arrest ↓ USP47 ↓ insulin-like growth factor 1 receptor, AKT, forkhead box O3α ↓ MAPK/Erk signaling pathway ↓ oncogenic characteristics ↓ PI3K/AKT/mTOR signal pathway ↓ FAK1-dependent signaling pathways ↑ E-cadherin protein ↓ TGFβ protein, TWIST1 gene ↓ STAT3	[303,306,310,313,314,315,316,317,319,320,322]
**Luteolin** 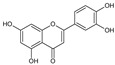	Found in flowers (*Reseda luteola and Chrysanthemums*), herbs (parsley, peppermint, oregano, and thyme), vegetables (celery seeds, onion leaves, cabbages, sweet bell peppers, carrots, and broccoli), and spices (cardamom and anise)	↓ DNA methyltransferases, some classical histone deacetylases, SIRT1 ↑ caspase-7, 8, 9 ↓ glutathione, IL-2, IFN-γ ↑ miR-124-3p, death receptor, MAPK ↑ caspase-3, Bax ↓ Bcl-2 ↑ chemo-sensitivity (5-FU) ↑ DNA double-strand breaks ↓ STAT3 ↓ TAM receptor tyrosine kinases ↑ Axl ↓ cyclin E, cyclin D ↓ Anoctamin 1 (ANO1) chloride channel activity ↓ Akt/mTOR/c-Myc signaling pathway	[331,336,340,341,343,346,347,348,349,350,352,353]

**Table 2 molecules-27-04818-t002:** The examples of clinical studies of natural products.

Natural Product	Experimental Design	Dosage	Comments	Reference
**Curcumin**	160 patients with solid tumor were given Meriva as an adjuvant to chemotherapy.	1500 mg/day of Meriva in three separate doses for six weeks.	Patients’ quality of life was improved, and systemic inflammation was significantly reduced.	[354]
	Patients with colorectal cancer were orally provided curcuma extract.	Up to 2.2 g daily (equal to 180 mg of curcumin) for several months.	Curcumin was proven to accumulate at the colorectum and acquire the effective therapeutic concentration.	[356]
	33 patients with benign prostatic hyperplasia were given Meriva.	1000 mg/day in two divided doses) for 2 years.	Improvements in all categories of the International Prostate Symptom Score.	[357]
	Patients with familial adenomatous polyposis were given a combination of curcumin and quercetin.	480 mg curcumin and 20 mg quercetin orally three times daily for six months.	The quantity and size of malignant polyps were dramatically reduced.	[359]
	25 patients with advanced pancreatic cancer were given curcumin capsules.	8 g of curcumin capsules daily, with restaging every eight weeks.	Oral curcumin had biological action and was safe in some pancreatic cancer patients.	[360]
**Resveratrol**	39 women with a high risk of breast cancer.	For 12 weeks, the participants were given two capsules per day containing either placebo, 5 mg of trans-resveratrol, or 50 mg of trans- resveratrol.	PGE2 levels were discovered to be reduced.	[365]
	Patients with colon cancer were given low-dose resveratrol and resveratrol-containing freeze-dried grape powder.	Low-dose resveratrol (80 mg/d) and resveratrol-containing freeze-dried grape powder (80 g/day for 14 days of treatment).	There was an increase in the expression of Myc and cyclin D1 in colon cancer tissue.	[367]
	Micronized resveratrol (SRT501) was given to individuals with colorectal cancer and hepatic metastases.	Micronized resveratrol (SRT501) was given at a dose of 5 g/day for two weeks.	SRT501 was well tolerated and increased mean plasma resveratrol levels (3.6-fold) after a single dosage compared to non-micronized resveratrol.	[369]
**EGCG**	481,563 volunteers aged 51–71 years were given hot tea.	1 cup/day or more of hot tea drinking for up to eight years of follow-up.	This study revealed a statistically significant inverse connection between hot tea drinking and risk of pharyngeal cancer.	[374]
	59 patients with oral mucosa leukoplakia were given green tea extract.	3 g mixed tea oral administration and topical treatment for six months.	37.9% of patients who received green tea treatment had smaller oral lesions.	[375]
	Assessment the relationship between green tea consumption and colorectal cancer risk on 69,710 Chinese women aged 40 to 70 years.	2–3 cups/day of green tea for up to 3 years of follow-up.	The study indicated that drinking tea on a regular basis considerably lowered the incidence of colorectal cancer.	[377]
	10 female patients (38–55 years old) with locally advanced noninflammatory breast cancer undergoing radiation were given EGCG capsules.	EGCG capsules (400 mg) were orally provided three times daily for two to eight weeks.	EGCG was discovered to increase the efficacy of radiotherapy in breast cancer patients, raising the possibility of EGCG being used as a therapeutic adjuvant in the treatment of metastatic breast cancer.	[378]
	42 patients with androgen-independent prostate cancer were given green tea.	6 g/day of green tea were provided orally in 6 divided doses for 2 months.	One of the patients demonstrated a 50% decrease in prostate-specific antigen (PSA) level.	[380]
	Patients with prostate cancer were prescribed green tea extract capsules.	Green tea extract capsules were given at a dose level of 250 mg twice daily for 2 months.	40% of patients who finished the treatment showed delayed disease progression.	[381]
	451 patients with pancreatic cancer were given green tea.	200 g/month of green tea were provided for 3 years.	The study lowered the risk of pancreatic cancer.	[383]
**Allicin**	51 patients with colorectal adenomas were given aged garlic extract.	2.4 mL/d of aged garlic extract for 12 months.	Aged garlic extract was related with a significantly lower risk of developing new colorectal adenomas.	[388]
	2526 persons with family history of stomach cancer were given allitridum and selenium.	200 mg synthetic allitridum every day and 100 microg selenium every other day for 2 years.	High dosages of allitridum and microdoses of selenium have been found to prevent stomach cancer, particularly in men.	[390]
	343 patients with esophageal squamous cell carcinoma and 755 cancer-free controls ingested raw garlic/onions.	Raw garlic/onions were given at least once per week for 10 years.	Raw onions/garlic were significantly protective against esophageal squamous cell carcinoma.	[393]
**Genistein**	9000 breast cancer patients were given soy food.	19.1 mg/day of soy food was given for up to 10 years.	Study found that increasing the genistein dose reduced breast cancer risk.	[396]
	23 prostate cancer patients received genistein before radical prostatectomy.	30 mg synthetic genistein was given daily for up to 6 weeks.	The level of the prostate cancer biomarker prostate-specific antigen in blood was reduced.	[398]

## Data Availability

Not applicable.

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
