# Peer review of "Plants as a Source of Anticancer Agents: From Bench to Bedside"

_molecules, 2022, doi:10.3390/molecules27154818_

Round 1

Reviewer 1 Report

The manuscript presented by Talib et al is a review on the potential anticancer activity of ten natural products. The authors present some mechanisms that contribute to cancer development and argue that a good strategy to fight cancer is to target multiple hallmarks. The manuscript deals with an interesting topic and it is well presented.

  Although the topic is of great interest, I was a little disappointed with the references used by the authors. They are not too old, but the use of plant natural products for cancer treatment, their mechanisms of action have been reported and reviewed in many papers. The authors should have concentrated in the most recent literature, to improve the impact of their work. I think that 2022 alone would provide data for a review with the same approach. However, despite the fact that it could be more updated, the manuscript is interesting.   The sizes of the chemical structures are too big in my opinion; I suggest reducing their size for aesthetical improvement.   Some specific suggestions are given below. Line 24: researchers (plural) Figure 1 is very interesting, however, the sharpness could be improved. Please, notice that there are two Figures 1. Please, check the abbreviations. Once you define them, use only the abbreviation; otherwise, why give the abbreviation? See, for example, EGF, defined in line 108, and again in line 162, 209 etc. Line 136. I understand that "Curcuma" is the genus, therefore, it should be in italics, even when you cite the genus but not the species.   Line 159. Please, add a full stop after [34] and start a new sentence. The same after [46], line 181. The authors sometimes use phrases too long; please, avoid for clarity. Line 182. curcumin, not Curcumin   LIne 192 and 193. trans and cis in italics, please. Line 194. Here the authors use the word "anti-oxidative" while in other parts they use "antioxidant". Unless they are referring to different activities, I advise to use only "antioxidant". Figure 2. I am uncomfortable with the representation of resveratrol chemical structure without defining the stereochemistry, since it is not unknown. How about presenting the structure of trans-resveratrol, which is the most stable form (calling attention to trans-resveratrol nomenclature in the heading of Figure 2)? Line 203. You gave the abbreviation of SOD and CAT. How about the other enzymes, is there an abbreviation for them?    About emodin (line 526). Is this metabolite toxic? It should be good to add a word on the toxicity -if there is toxicity-, to give a better idea of the metabolite limitations, instead of presenting only the positive data. About quercetin, gallte, etc (apply to all metabolites): Can you correlate the different mechanisms to Figure 1? It would be helpful to the readers to have an idea how the different mechanisms described to the different metabolites correlate to the main activities represented in the Figure. I suggest numbering the biomarkers in Figure 1 (viz CD8... = A; ILs... = B, etc) and, when mentioning them in the text, add (see Fig1 A, etc).  Figure 4. The stereochemistry of one of the aromatic rings is not good. As it is, it seems that there is a bond out of the plane in the ring, which is not possible. You can correct that by showing the stereochemistry of the hydrogen linked to the heterocyclic ring and leaving plain the bond between the heterocycle and the aromatic ring. Line 416. Please, correct Li et al. Line 432. Please, correct Jiang Table 1. I don't understand why the references were grouped or given separately. If different groups refer to different actions, the table must be formatted correctly.   In some parts of the manuscript (for example, lines 858-885), the authors seem to present a series of information, each one in a different paragraph. It is necessary to link the different points instead of given fragmented information.

Author Response

We would like to thank you for your time and the productive comments and suggestions. The manuscript was revised and corrected as requested with a detailed response for each comment (please see attached).

We hope that our revised manuscript will meet your expectations 

Reviewer 2 Report

Dear Authors that submitted the Review Ms Plants As a Source of Anticancer Agents: from Bench to Bed Side. By Wamidh H. Talib1, Safa Daoud, Asma Ismail Mahmod, Reem Ali Hamed, Dima Awajan, Sara Feras Abuarab, Lena Hisham Odeh, Samar Khater and Lina T. Al Kury. For possible publication at the Journal. After reviewing In Introduction say that: Cancer is a chronic disease and one of the main causes of death around the world. Phytochemicals, a bioactive compounds have diverse therapeutic potentials, including anti- diabetic, anti-inflammatory, cardiovascular protective, antioxidant, and anticancer effects. Plant secondary metabolites are as flavonoids, alkaloids, phytosterols, terpenoids, sulfides, polyphenols, and others have been considered an important reservoir for novel anticancer agents. They have properties such as tumor growth inhibition, apoptosis induction, immune modulation, and angiogenesis suppression, which make them suitable candidates for anticancer drug development. The aim of the Ms is to review ten plant-derived natural products with anticancer properties describing their chemical structures, natural sources, general biological activities, and explain thoroughly their anti-carcinogenic activity. The majority of these natural products inhibit cancer by targeting multiple cancer hallmarks and many of them reached clinical applications. However, Clinical studies of some natural product like Quercetin, Thymoquinone, Emodin, Parthenolide, Luteolin need to be investigated. Apoptosis induction is the most common pathway activated by plant derived natural products. The authors reviewed 323 articles. The content of the review is interesting for the Journal and audience. Therefore I suggest publishing at the Journal.I suggest publishing at the Journal.I suggest publishing at the Journal. It’s well written, very easy to understand. Sincerely yours, Blas Lotina Hennsen, PhD.

Author Response

Thank you very much for your positive feedback and constructive comments. The article was revised and corrected as requested.

We hope that our revised manuscript will meet your expectations 

Reviewer 3 Report

The authors present an interesting study on the protective effects of some plants against cancer development. This paper is well written and the organization of the paper is also adequate.  The figures and tables are comprehensive and support the content. However, there are various sections that require revision to improve the readability of the paper.

Specific comment:

Throughout the manuscript, various hyperlinks are given that are not appreciable and must be removed from the manuscript, for example, on page 7, line 245 (Querus), page 8 line 288 (apoptosis), 295, 297, 298, 310, 311, 315, 316, etc. I would suggest scrutinizing the manuscript for the same.

General comments

1.    The manuscript requires extensive linguistic and grammatical revision.

2.    The prevalence rate and global burden of cancer should be included in the introduction section.

3.    Since there is little to no real separation of the introduction’s ideas into focused paragraphs, it can be difficult to determine which concepts are the most crucial to move forward with until the reader has read further into the manuscript.

4.    The introduction is poorly explained and could use better or more in-depth explanations of the topics mentioned, such as describing the effects of the secondary metabolites in cancer (given that the concept was deemed important enough to mention or to begin with).

5.    It is not clear why only these 10 phytochemicals were chosen.

6.    Cell cycle target inhibition and anti-cancer drug discovery should be incorporated into the manuscript.

7.    It is advisable to incorporate proper mechanisms involved in cancer chemoprevention and treatment in order to enhance the readability and understating., if possible.

8.    As throughout the manuscript, uniformity and connection between the paragraphs are missing.

9.    It could be interesting if the authors incorporate these chemical structures in table 1 rather than in the manuscript.

10. In the references, most of the citations must be recent (not older than 5 years), verify the reference section for the same.

Author Response

We would like to thank you for your time and productive comments. The manuscript was revised and corrected as requested in your comments. A detailed response was also prepared to explain each correction (Please see attached).

We hope that the revised manuscript will meet your expectations.
